# PARALLEL DEEP NEURAL NETWORKS
# HAVE ZERO DUALITY GAP

## ABSTRACT

Training deep neural networks is a well-known highly non-convex problem. In recent work (Pilanci & Ergen, 2020), it is shown that there is no duality gap for regularized two-layer neural networks with ReLU activation, which enables global optimization via convex programs. For multi-layer linear networks with vector outputs, we formulate convex dual problems and demonstrate that the duality gap is non-zero for depth three and deeper networks. However, by modifying the deep networks to more powerful parallel architectures, we show that the duality gap is exactly zero. Therefore, strong convex duality holds, and hence there exist equivalent convex programs that enable training deep networks to global optimality. We also demonstrate that the weight decay regularization in the parameters explicitly encourages low-rank solutions via closed-form expressions. For three-layer non-parallel ReLU networks, we show that strong duality holds for rank-1 data matrices, however, the duality gap is non-zero for whitened data matrices. Similarly, by transforming the neural network architecture into a corresponding parallel version, the duality gap vanishes.

## 1 INTRODUCTION

Deep neural networks demonstrate outstanding representation and generalization abilities in popular learning problems ranging from computer vision, natural language processing to recommendation system. Although the training problem of deep neural networks is a highly non-convex optimization problem, simple algorithms, such as stochastic gradient descent, can find a solution with good generalization properties. The non-convex and non-linear nature of neural networks render the theoretical understanding of neural networks extremely challenging.

The Lagrangian dual problem (Boyd et al., 2004) plays an important role in the theory of convex and non-convex optimization. For convex optimization problems, the convex duality is an important tool to determine its optimal value and to characterize the optimal solutions. Even for a non-convex primal problem, the dual problem is a convex optimization problem the can be solved efficiently. As a result of weak duality, the optimal value of the dual problem serves as a non-trivial lower bound for the optimal primal objective value. Although the duality gap is non-zero for non-convex problems, the dual problem provides a convex relaxation of the non-convex primal problem. For example, the semi-definite programming relaxation of the two-way partitioning problem can be derived from its dual problem Boyd et al. (2004).

The convex duality also has important applications in machine learning. In (Paternain et al., 2019), the design problem of an all-encompassing reward can be formulated as a constrained reinforcement learning problem, which is shown to have zero duality. This property gives a theoretical convergence guarantee of the primal-dual algorithm for solving this problem. Meanwhile, the minimax generative adversarial net (GAN) training problem can be tackled using duality (Farnia & Tse, 2018).

In lines of recent works, the convex duality can also be applied for analyzing the optimal layer weights of two-layer neural networks with linear or ReLU activations (Pilanci & Ergen, 2020; Ergen & Pilanci, 2020a). Based on the convex duality framework, the training problem of two-layer neural networks with ReLU activation can be represented in terms of a single convex program in (Pilanci & Ergen, 2020). Such convex optimization formulations are extended to two-layer and three-layer convolutional neural network training problems in (Ergen & Pilanci, 2020c). Strong duality also holds for deep linear neural networks with scalar output (Ergen & Pilanci, 2020b). The convex

optimization formulation essentially gives a detailed characterization of the global optimum of the training problem. This enables us to examine in numerical experiments whether popular optimizers for neural networks, such as gradient descent or stochastic gradient descent, converge to the global optimum of the training loss.

Admittedly, the zero duality gap is hard to achieve for deep neural networks, especially for those with vector outputs. This imposes more difficulty to train deep neural networks. Fortunately, neural networks with parallel structures (also known as multi-branch architecture) appear to be easier to train. Practically, the usage of parallel neural networks dates back to AlexNet (Krizhevsky et al., 2012). Modern neural network architecture including Inception (Szegedy et al., 2017), Xception (Chollet, 2017) and SqueezeNet (Iandola et al., 2016) utilize the parallel structure. As the "parallel" version of ResNet (He et al., 2016a;b), ResNeXt (Xie et al., 2017) and Wide ResNet (Zagoruyko & Komodakis, 2016) exhibit improved performance on many applications. Recently, it was shown that neural networks with the parallel architecture have smaller duality gaps (Zhang et al., 2019) compared to standard neural networks. On the other hand, it is known that overparameterized parallel neural networks have benign training landscapes (Haeffele & Vidal, 2017). For training $\ell_2$ loss with deep linear networks using Schatten norm regularization, Zhang et al. (2019) show that there is no duality gap. From another perspective, the standard two-layer network is equivalent to the parallel two-layer network. This may also explain why there is no duality gap for two-layer neural networks.

## 1.1 CONTRIBUTIONS

Following the convex duality framework introduced in (Ergen & Pilanci, 2020b;a), which showed the duality gap is zero for two-layer networks, we go beyond two-layer and study the convex duality for vector-output deep neural networks with linear activation and ReLU activation. **Surprisingly, we prove that three-layer networks may have duality gaps depending on their architecture, unlike two-layer neural networks which always have zero duality gap**. We summarize our contributions as follows.

- For training standard vector-output deep linear networks using $\ell_2$ regularization, we precisely calculate the optimal value of the primal and dual problems and show that the **duality gap is non-zero**, i.e., Lagrangian relaxation is inexact. We also demonstrate that the $\ell_2$-regularization on the parameter explicitly forces a tendency toward a low-rank solution, which is boosted with the depth. However, we show that the optimal solution is available in **closed-form**.

- For parallel deep linear networks, with certain convex regularization, we show that **the duality gap is zero**, i.e, Lagrangian relaxation is exact.

- For training vector-output three-layer ReLU networks with standard architecture using $\ell_2$ regularization, even when the data is whitened, we show that the **duality gap is non-zero**. The gap can be closed by replacing the neural network to one with parallel architecture.

- For parallel deep ReLU networks of arbitrary depth, with certain convex regularization, we prove strong duality, i.e., **the duality gap is zero**. Remarkably, this guarantees that **there is a convex program equivalent to the original deep ReLU neural network problem.**

We summarize the duality gaps for parallel/standard neural network in Table 1.

## 1.2 NOTATIONS

We use bold capital letters to represent matrices and bold lowercase letters to represent vectors. Denote $[n] = \{1, \ldots, n\}$. For a matrix $\mathbf{W}_l \in \mathbb{R}^{m_1 \times m_2}$, for $i \in [m_1]$ and $j \in [m_2]$, we denote $\mathbf{w}_{l,i}^{\text{col}}$ as its $i$-th column and $\mathbf{w}_{l,j}^{\text{row}}$ as its $j$-th row. Throughout the paper, $\mathbf{X} \in \mathbb{R}^{N \times d}$ is the data matrix consisting of $d$ dimensional $N$ samples and $\mathbf{Y} \in \mathbb{R}^{N \times K}$ is the label matrix for a regression/classification task with $K$ outputs.

## 1.3 MOTIVATIONS AND BACKGROUND

Recently a series of papers (Pilanci & Ergen, 2020; Ergen & Pilanci, 2020b;a) studied two-layer neural networks via convex duality and proved that strong duality holds for these architectures.

Table 1: Duality gaps for $L$-layer standard and parallel architectures. we compare our duality gap characterization with previous literature. Each check mark indicates whether a characterization of the duality gap exists for the corresponding architecture and the number next to it indicates whether the gap is zero or not.

| | | Linear | | | ReLU | | |
|---|---|---|---|---|---|---|---|
| | | $L=2$ | $L=3$ | $L>3$ | $L=2$ | $L=3$ | $L>3$ |
| **Standard** | Ergen & Pilanci (2020a; 2021a) | ✗ | ✗ | ✗ | ✓(0) | ✗ | ✗ |
| | Bach (2017) Pilanci & Ergen (2020) | ✓(0) | ✗ | ✗ | ✓(0) | ✗ | ✗ |
| | Ergen & Pilanci (2021b) | ✗ | ✗ | ✗ | ✓(0) | ✗ | ✗ |
| | Ergen & Pilanci (2021c) | ✓(0) | ✗ | ✗ | ✗ | ✗ | ✗ |
| | **This paper** | ✓(0) | ✓($\neq 0$) | ✓($\neq 0$) | ✓(0) | ✓($\neq 0$) | ✗ |
| **Parallel** | Ergen & Pilanci (2021b) | ✗ | ✗ | ✗ | ✓(0) | ✓(0) | ✗ |
| | Zhang et al. (2019) Ergen & Pilanci (2021c) | ✓(0) | ✓(0) | ✓(0) | ✗ | ✗ | ✗ |
| | **This paper** | ✓(0) | ✓(0) | ✓(0) | ✓(0) | ✓(0) | ✓(0) |

Particularly, these prior works consider the following weight decay regularized training framework for classification/regression tasks. Given a data matrix $\mathbf{X} \in \mathbb{R}^{N \times d}$ consisting of $d$ dimensional $N$ samples and the corresponding label matrix $\mathbf{y} \in \mathbb{R}^N$, the weight-decay regularized training problem for a scalar-output neural network with $m$ hidden neurons can be written as follows

$$P := \min_{\mathbf{W}_1, \mathbf{w}_2} \frac{1}{2} \|\phi(\mathbf{X}\mathbf{W}_1)\mathbf{w}_2 - \mathbf{y}\|_2^2 + \frac{\beta}{2}(\|\mathbf{W}_1\|_F^2 + \|\mathbf{w}_2\|_2^2), \tag{1}$$

where $\mathbf{W}_1 \in \mathbb{R}^{d \times m}$ and $\mathbf{w}_2 \in \mathbb{R}^m$ are the layer weights, $\beta > 0$ is a regularization parameter, and $\phi$ is the activation function, which can be linear $\phi(z) = z$ or ReLU $\phi(z) = \max\{z, 0\}$. Then, one can take the dual of (1) with respect to $\mathbf{W}_1$ and $\mathbf{w}_2$ obtain the following dual optimization problem

$$D := \max_{\boldsymbol{\lambda}} -\frac{1}{2}\|\boldsymbol{\lambda} - \mathbf{y}\|_2^2 + \frac{1}{2}\|\mathbf{y}\|_2^2, \text{ s.t. } \max_{\mathbf{w}_1 : \|\mathbf{w}_1\|_2 \leq 1} |\boldsymbol{\lambda}^T \phi(\mathbf{X}\mathbf{w}_1)| \leq \beta. \tag{2}$$

We first note that since the training problem (1) is non-convex, strong duality may not hold, i.e., $P \geq D$. Surprisingly, as shown in Pilanci & Ergen (2020); Ergen & Pilanci (2020b;a), strong duality in fact holds, i.e., $P = D$, for two-layer networks and therefore one can derive exact convex representations for the non-convex training problem in (1). However, extensions of this approach to deeper and state-of-the-art architectures are not available in the literature. Based on this observation, the central question we address in this paper is:

*Does strong duality hold for deep neural networks?*

Depending on the answer to the question above, an immediate next questions we address is

*Can we characterize the duality gap (P-D)? Is there an architecture for which strong duality holds independent of the depth?*

Consequently, throughout the paper, we provide a full characterization of convex duality for deeper neural networks. Then, based on this characterization, we propose a modified architecture for which strong duality holds regardless of depth.

### 1.4 CONVEX DUALITY FOR TWO-LAYER NEURAL NETWORKS

We briefly review the convex duality for two-layer neural networks introduced in (Ergen & Pilanci, 2020b;a). Consider the following weight-decay regularized training problem for a vector-output neural network architecture with $m$ hidden neurons

$$\min_{\mathbf{W}_1, \mathbf{W}_2} \frac{1}{2} \|\phi(\mathbf{X}\mathbf{W}_1)\mathbf{W}_2 - \mathbf{Y}\|_F^2 + \frac{\beta}{2}(\|\mathbf{W}_1\|_F^2 + \|\mathbf{W}_2\|_F^2), \tag{3}$$

where $\mathbf{W}_1 \in \mathbb{R}^{d \times m}$ and $\mathbf{W}_2 \in \mathbb{R}^{m \times K}$ are the variables, and $\beta > 0$ is a regularization parameter. Here $\phi$ is the activation function, which can be linear $\phi(z) = z$ or ReLU $\phi(z) = \max\{z, 0\}$. As long as the network is sufficiently overparameterized, there exists a feasible solution for such that $\phi(\mathbf{X}\mathbf{W}_1)\mathbf{W}_2 = \mathbf{Y}$. Then, a minimal norm variant[1] of the training problem in (3) is given by

$$\min_{\mathbf{W}_1, \mathbf{W}_2} \frac{1}{2}(\|\mathbf{W}_1\|_F^2 + \|\mathbf{W}_2\|_F^2) \text{ s.t. } \phi(\mathbf{X}\mathbf{W}_1)\mathbf{W}_2 = \mathbf{Y}. \tag{4}$$

As shown in (Pilanci & Ergen, 2020), after a suitable rescaling, this problem can be reformulated as

$$\min_{\mathbf{W}_1, \mathbf{W}_2} \sum_{j=1}^{m} \|\mathbf{w}_{2,j}^{\text{row}}\|_2 \text{ s.t. } \phi(\mathbf{X}\mathbf{W}_1)\mathbf{W}_2 = \mathbf{Y}, \|\mathbf{w}_{1,j}^{\text{col}}\|_2 \leq 1, j \in [m]. \tag{5}$$

where $[m] = \{1, \ldots, m\}$. Here $\mathbf{w}_{2,j}^{\text{row}}$ represents the $j$-th row of $\mathbf{W}_2$ and $\mathbf{w}_{1,j}^{\text{col}}$ denotes the $j$-th column of $\mathbf{W}_1$. The rescaling does not change the solution to (4). By taking the dual with respect to $\mathbf{W}_1$ and $\mathbf{W}_2$, the dual problem of (5) with respect to variables is a convex optimization problem given by

$$\max_{\mathbf{\Lambda}} \text{tr}(\mathbf{\Lambda}^T \mathbf{Y}), \text{ s.t. } \max_{\mathbf{u}: \|\mathbf{u}\|_2 \leq 1} \|\mathbf{\Lambda}^T \phi(\mathbf{X}\mathbf{u})\|_2 \leq 1, \tag{6}$$

where $\mathbf{\Lambda} \in \mathbb{R}^{N \times K}$ is the dual variable. Provided that $m \geq m^*$, where $m^* \leq N + 1$, the strong duality holds, i.e., the optimal value of the primal problem (5) equals to the optimal value of the dual problem (6).

## 1.5 Organization

This paper is organized as follows. In Section 2, we review standard neural networks and introduce parallel architectures. For deep linear networks, we derive primal and dual problems for both standard and parallel architectures and provide calculations of optimal values of these problems in Section 3. We derive primal and dual problems for three-layer ReLU networks with standard architecture and precisely calculate the optimal values for whitened data in Section 4. We also show that deep ReLU networks with parallel structures have no duality gap.

## 2 Standard neural networks vs parallel architectures

We start with the $L$-layer neural network with the standard architecture:

$$f_{\boldsymbol{\theta}}(\mathbf{X}) = \mathbf{A}_{L-1}\mathbf{W}_L, \quad \mathbf{A}_l = \phi(\mathbf{A}_{l-1}\mathbf{W}_l), \forall l \in [L-1], \mathbf{A}_0 = \mathbf{X},$$

where $\phi$ is the activation function, $\mathbf{W}_l \in \mathbb{R}^{m_{l-1} \times m_l}$ is the weight matrix in the $l$-th layer and $\theta = (\mathbf{W}_1, \ldots, \mathbf{W}_L)$ represents the parameter of the neural network. We also denote the input and output dimensions as $m_0 = d$ and $m_L = K$ for simplicity and assume that $m_l \geq \max\{d, K\}, \forall l \in [L-2]$.

We then introduce the neural network with the parallel architecture:

$$f_{\boldsymbol{\theta}}^{\text{prl}}(\mathbf{X}) = \mathbf{A}_{L-1}\mathbf{W}_L,$$
$$\mathbf{A}_{l,j} = \phi(\mathbf{A}_{l-1,j}\mathbf{W}_{l,j}), \forall j \in [m], \forall l \in [L-1],$$
$$\mathbf{A}_{0,j} = \mathbf{X}, \forall j \in [m].$$

Here for $l \in [L-1]$, the $l$-th layer has $m$ weight matrices $\mathbf{W}_{l,j} \in \mathbb{R}^{m_{l-1} \times m_l}$ where $j \in [m]$. Specifically, we let $m_{L-1} = 1$. In short, we can view the output $\mathbf{A}_{L-1}$ from a parallel neural network as a concatenation of $m$ scalar-output standard neural work. In Figure 1 and 2, we provide examples of neural networks with standard and parallel architectures. We shall emphasize that for $L = 2$, the standard neural network is identical to the parallel neural network.

---

[1]This corresponds to weak regularization, i.e., $\beta \to 0$ in (3) as considered in Wei et al. (2018).

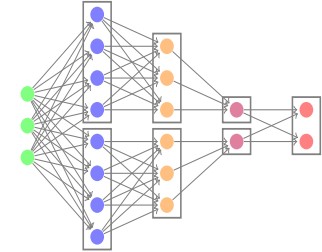

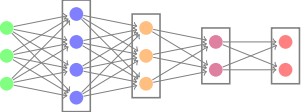

Figure 1: Standard Architecture          Figure 2: Parallel Architecture

## 3 DEEP LINEAR NETWORKS

In this section, we discuss the convex duality of deep linear network with vector output.

### 3.1 STANDARD DEEP LINEAR NETWORKS

We first consider the neural network with standard architecture, i.e., $f_{\boldsymbol{\theta}}(\mathbf{X}) = \mathbf{X}\mathbf{W}_1 \ldots \mathbf{W}_L$. Consider the following minimal norm optimization problem:

$$P_{\mathrm{lin}} = \min_{\{\mathbf{W}_l\}_{l=1}^L} \frac{1}{2} \sum_{i=1}^L \|\mathbf{W}_l\|_F^2, \text{ s.t. } \mathbf{X}\mathbf{W}_1, \ldots, \mathbf{W}_L = \mathbf{Y}, \tag{7}$$

where the variables are $\mathbf{W}_1, \ldots, \mathbf{W}_L$. As shown in the Proposition 3.1 in (Ergen & Pilanci, 2020b), by introducing a scale parameter $t$, the problem (7) can be reformulated as

$$P_{\mathrm{lin}} = \min_{t>0} \frac{L-2}{2} t^2 + P_{\mathrm{lin}}(t),$$

where $P_{\mathrm{lin}}(t)$ is defined as

$$
\begin{aligned}
P_{\mathrm{lin}}(t) = \min_{\{\mathbf{W}_l\}_{l=1}^L} &\sum_{j=1}^K \|\mathbf{w}_{L,j}^{\mathrm{row}}\|_2, \\
\text{s.t. } &\|\mathbf{W}_i\|_F \leq t, i \in [L-2], \|\mathbf{w}_{L-1,j}^{\mathrm{col}}\|_2 \leq 1, j \in [m_{L-1}], \\
&\mathbf{X}\mathbf{W}_1 \ldots \mathbf{W}_L = \mathbf{Y}.
\end{aligned}
\tag{8}
$$

The following proposition characterize the dual problem of $P_{\mathrm{lin}}(t)$ and its bi-dual, i.e., dual of the dual problem.

**Proposition 1** *The dual problem of $P_{\mathrm{lin}}(t)$ defined in* (8) *is a convex optimization problem given by*

$$
\begin{aligned}
D_{\mathrm{lin}}(t) = \max_{\boldsymbol{\Lambda}} &\operatorname{tr}(\boldsymbol{\Lambda}^T \mathbf{Y}) \\
\text{s.t. } &\|\boldsymbol{\Lambda}^T \mathbf{X}\mathbf{W}_1 \ldots \mathbf{W}_{L-2}\mathbf{w}_{L-1}\|_2 \leq 1, \\
&\|\mathbf{W}_i\|_F \leq t, \forall i \in [L-2], \|\mathbf{w}_{L-1}\|_2 \leq 1.
\end{aligned}
\tag{9}
$$

*There exists $m^* \leq KN + 1$ such that the dual problem $D_{\mathrm{lin}}(t)$ can be reformulated as the bi-dual, i.e.,*

$$
\begin{aligned}
D_{\mathrm{lin}}(t) = \min_{\{\mathbf{W}_{l,j}\}_{l=1}^L} &\sum_{j=1}^{m^*} \|\mathbf{w}_{L,j}^{\mathrm{row}}\|_2, \\
\text{s.t. } &\sum_{j=1}^{m^*} \mathbf{X}\mathbf{W}_{1,j} \ldots \mathbf{W}_{L-2,j}\mathbf{w}_{L-1,j}^{\mathrm{col}}\mathbf{w}_{L,j}^{\mathrm{row}} = \mathbf{Y}, \\
&\|\mathbf{W}_{i,j}\|_F \leq t, i \in [L-2], j \in [m^*], \|\mathbf{w}_{L-1,j}^{\mathrm{col}}\|_2 \leq 1, j \in [m^*].
\end{aligned}
\tag{10}
$$

Detailed derivation of the dual and the bi-dual are provided in Appendix B.1. The reason why we do not directly take the dual of $P_{\text{lin}}$ is that the objective function in $P_{\text{lin}}$ involves the weights of first $L-1$ layer, which prevents obtaining a meaningful dual problem. We note that bi-dual problem is related to the minimal norm problem of a parallel neural network with aligned weights. Namely, the Frobenius norm of the weight matrices in each branch has the same upper bound $t$.

For a matrix $\mathbf{A} \in \mathbb{R}^{m \times n}$ and $p > 0$, the Schatten-$p$ quasi-norm of $\mathbf{A}$ is defined as

$$\|\mathbf{A}\|_{S_p} = \left( \sum_{i=1}^{\min\{m,n\}} \sigma_i^p(\mathbf{A}) \right)^{1/p},$$

where $\sigma_i(\mathbf{A})$ is the $i$-th largest singular value of $\mathbf{A}$. The optimal value of $P_{\text{lin}}(t)$ and $D_{\text{lin}}(t)$ can be precisely calculated.

**Theorem 1** *For fixed $t > 0$, the optimal value of $P_{\text{lin}}(t)$ and $D_{\text{lin}}(t)$ are given by*

$$P_{\text{lin}}(t) = t^{-(L-2)} \|\mathbf{X}^\dagger \mathbf{Y}\|_{S_{2/L}}, \tag{11}$$

*and*

$$D_{\text{lin}}(t) = t^{-(L-2)} \|\mathbf{X}^\dagger \mathbf{Y}\|_*, \tag{12}$$

*where $\mathbf{X}^\dagger$ is the pseudo inverse of $\mathbf{X}$ and $\|\cdot\|_*$ represents the nuclear norm.*

As a result, the duality gap exists, i.e., $P > D$, for standard deep linear networks with $L \geq 3$, if the singular values of $X^\dagger Y$ are not equal to the same value. We note that the optimal scale parameter $t$ for the primal problem $P_{\text{lin}}$ is given by $t^* = \|\mathbf{W}^*\|_{S_{2/L}}^{1/L}$. To the best of our knowledge, this result wasn't shown previously. We leave the proof in Appendix B.2 and provide conditions on $\mathbf{W}_1, \ldots, \mathbf{W}_L$ to achieve the optimal value.

In the calculation of $P_{\text{lin}}(t)$, we utilize the following proposition.

**Proposition 2** *Suppose that $\mathbf{W} \in \mathbb{R}^{d \times K}$ with rank $r$ is given. Assume that $m_i \geq r$ for $i = 0, \ldots, L$. Consider the following optimization problem:*

$$\min_{\{\mathbf{W}_l\}_{l=1}^L} \frac{1}{2} \left( \|\mathbf{W}_1\|_F^2 + \cdots + \|\mathbf{W}_L\|_F^2 \right), \ s.t. \ \mathbf{W}_1 \mathbf{W}_2 \ldots \mathbf{W}_L = \mathbf{W}. \tag{13}$$

*Then, the optimal value of the problem* (13) *is given by $\frac{L}{2}\|\mathbf{W}\|_{S_{2/L}}^{2/L}$.*

### 3.2 PARALLEL DEEP LINEAR NEURAL NETWORKS

For the neural network with parallel structure, we consider the minimal norm optimization problem:

$$\min_{\{\mathbf{W}_{l,j}\}_{l=1}^L} \frac{1}{2} \left( \sum_{l=1}^{L-1} \sum_{j=1}^m \|\mathbf{W}_{l,j}\|_F^2 + \|\mathbf{W}_L\|_F^2 \right),$$

$$\text{s.t.} \ \sum_{j=1}^m \mathbf{X} \mathbf{W}_{1,j} \ldots \mathbf{W}_{L-2,j} \mathbf{w}_{L-1,j}^{\text{col}} \mathbf{w}_{L,j}^{\text{row}} = \mathbf{Y}. \tag{14}$$

Due to a rescaling to achieve the lower bound of the inequality of arithmetic and geometric means, we have the following result.

**Proposition 3** *The problem* (14) *can be formulated as*

$$\min_{\{\mathbf{W}_{l,j}\}_{l \in [L], j \in [m]}} \frac{L}{2} \sum_{j=1}^m \|\mathbf{w}_{L,j}^{\text{row}}\|_2^{2/L},$$

$$\text{s.t.} \ \sum_{j=1}^m \mathbf{X} \mathbf{W}_{1,j} \ldots \mathbf{W}_{L-2,j} \mathbf{w}_{L-1,j}^{\text{col}} \mathbf{w}_{L,j}^{\text{row}} = \mathbf{Y}, \tag{15}$$

$$\|\mathbf{W}_{l,j}\|_F \leq 1, l \in [L-2], j \in [m], \|\mathbf{w}_{L-1,j}^{\text{col}}\|_2 \leq 1, j \in [m].$$

We note that $z^{2/L}$ is a non-convex of $z$ and we cannot take a meaningful dual. The bi-dual problem of the standard neural network does NOT correspond to the primal problem of the parallel neural network because the Frobenius norm of each parallel part of the parallel architecture can be different in formulating (15).

We can consider another regularization, i.e.,

$$
\begin{aligned}
P_{\text{lin}}^{\text{prl}} = \min_{\{\mathbf{W}_{l,j}\}_{l=1}^{L}} \quad & \frac{1}{2} \sum_{l=1}^{L-1} \sum_{j=1}^{m} \|\mathbf{W}_{l,j}\|_F^L + \|\mathbf{W}_L\|_F^L, \\
\text{s.t.} \quad & \sum_{j=1}^{m} \mathbf{X}\mathbf{W}_{1,j}\ldots\mathbf{W}_{L-2,j}\mathbf{w}_{L-1,j}^{\text{col}}\mathbf{w}_{L,j}^{\text{row}} = \mathbf{Y}.
\end{aligned}
\tag{16}
$$

**Proposition 4** *The problem* (16) *can be formulated as*

$$
\begin{aligned}
P_{\text{lin}}^{\text{prl}} = \min_{\{\mathbf{W}_{l,j}\}_{l\in[L], j\in[m]}} \quad & \frac{L}{2} \sum_{j=1}^{m} \|\mathbf{w}_{L,j}^{\text{row}}\|_2, \\
\text{s.t.} \quad & \sum_{j=1}^{m} \mathbf{X}\mathbf{W}_{1,j}\ldots\mathbf{W}_{L-2,j}\mathbf{w}_{L-1,j}^{\text{col}}\mathbf{w}_{L,j}^{\text{row}} = \mathbf{Y}, \\
& \|\mathbf{W}_{l,j}\|_F \le 1, l \in [L-2], j \in [m], \|\mathbf{w}_{L-1,j}^{\text{col}}\|_2 \le 1, j \in [m].
\end{aligned}
\tag{17}
$$

*The dual problem of $P_{\text{lin}}^{\text{prl}}$ is a convex problem*

$$
\begin{aligned}
D_{\text{lin}}^{\text{prl}} = \max_{\mathbf{\Lambda}} \quad & \text{tr}(\mathbf{\Lambda}^T \mathbf{Y}), \\
\text{s.t.} \quad & \|\mathbf{\Lambda}^T \mathbf{X}\mathbf{W}_1\ldots\mathbf{W}_{L-2}\mathbf{w}_{L-1}\|_2 \le L/2, \\
& \forall \|\mathbf{W}_i\|_F \le 1, i \in [L-2], \|\mathbf{w}_{L-1}\|_2 \le 1.
\end{aligned}
\tag{18}
$$

For the parallel linear network, the strong duality holds.

**Theorem 2** *There exists $m^* \le KN+1$ such that as long as the number of branches $m \ge m^*$, the strong duality holds for the problem* (16). *Namely, $P_{\text{lin}}^{\text{prl}} = D_{\text{lin}}^{\text{prl}}$. The optimal values are both $\frac{L}{2}\|\mathbf{X}^\dagger \mathbf{Y}\|_*$.*

This implies that there exist equivalent convex problems which achieve the global optimum of the deep parallel linear network. Comparatively, optimizing deep parallel linear neural networks can be much easier than optimizing deep standard linear networks.

### 3.3 GENERAL LOSS FUNCTIONS

Now we consider general loss functions, i.e.,

$$
\min_{\{\mathbf{W}_l\}_{l=1}^{L}} \ell(\mathbf{X}\mathbf{W}_1\ldots\mathbf{W}_L, \mathbf{Y}) + \frac{\beta}{2} \sum_{i=1}^{L} \|\mathbf{W}_i\|_F^2,
$$

where $\ell(\mathbf{Z}, \mathbf{Y})$ is a general loss function and $\beta > 0$ is a regularization parameter. According to Proposition 2, the above problem is equivalent to

$$
\min_{\mathbf{W}} \ell(\mathbf{X}\mathbf{W}, \mathbf{Y}) + \frac{\beta L}{2} \|\mathbf{W}\|_{S_{2/L}}^{2/L}.
\tag{19}
$$

The $\ell_2$ regularization term becomes the Schatten-$2/L$ quasi-norm on $\mathbf{W}$ to the power $2/L$. Suppose that there exists $\mathbf{W}$ such that $l(\mathbf{X}\mathbf{W}, \mathbf{Y}) = 0$. With $\beta \to 0$, asymptotically, the optimal solution to the problem (19) converges to the optimal solution of

$$
\min_{\mathbf{W}} \|\mathbf{W}\|_{S_{2/L}}^{2/L}, \text{ s.t. } \ell(\mathbf{X}\mathbf{W}, \mathbf{Y}) = 0.
\tag{20}
$$

In other words, the $\ell_2$ regularization explicitly regularizes the training problem to find a low-rank solution $\mathbf{W}$.

## 4 Neural networks with ReLU activation

Now, we focus on the three-layer neural network with ReLU activation, i.e., $\phi(z) = \max\{z, 0\}$.

### 4.1 Standard three-layer ReLU networks

We first focus on the three-layer ReLU network with standard architecture. Consider the minimal norm problem

$$P_{\text{ReLU}} = \min_{\{\mathbf{W}_i\}_{i=1}^3} \frac{1}{2} \sum_{i=1}^3 \|\mathbf{W}_i\|_F^2, \text{ s.t. } ((\mathbf{X}\mathbf{W}_1)_+\mathbf{W}_2)_+\mathbf{W}_3 = \mathbf{Y}. \tag{21}$$

Here we denote $(z)_+ = \max\{z, 0\}$. Similarly, by introducing a scale parameter $t$, this problem can be formulated as

$$P_{\text{ReLU}} = \min_{t>0} \frac{1}{2} t^2 + P_{\text{ReLU}}(t),$$

where $P_{\text{ReLU}}(t)$ is defined as

$$P_{\text{ReLU}}(t) = \min_{\{\mathbf{W}_i\}_{i=1}^3} \sum_{j=1}^K \|\mathbf{w}_{3,j}^{\text{row}}\|_2,$$
$$\text{s.t. } \|\mathbf{W}_1\|_F \le t, \|\mathbf{w}_{2,j}^{\text{col}}\|_2 \le 1, j \in [m_2], \tag{22}$$
$$((\mathbf{X}\mathbf{W}_1)_+\mathbf{W}_2)_+\mathbf{W}_3 = \mathbf{Y}.$$

The proof is analogous to the proof of Proposition 3.1 in (Ergen & Pilanci, 2020b). For $\mathbf{W}_1 \in \mathbb{R}^{d \times m}$, we define the set

$$\mathcal{A}(\mathbf{W}_1) = \{((\mathbf{X}\mathbf{W}_1)_+\mathbf{w}_2)_+ | \|\mathbf{w}_2\|_2 \le 1\}. \tag{23}$$

**Proposition 5** *The dual problem of $P_{\text{ReLU}}(t)$ defined in (22) is a convex problem defined as*

$$D_{\text{ReLU}}(t) = \max_{\mathbf{\Lambda}} \text{tr}(\mathbf{\Lambda}^T \mathbf{Y}), \text{ s.t. } \|\mathbf{\Lambda}^T \mathbf{v}\|_2 \le 1, \mathbf{v} \in \mathcal{A}(\mathbf{W}_1), \forall \|\mathbf{W}_1\|_F \le t. \tag{24}$$

*There exists $m^* \le KN + 1$ such that the dual problem can be reformulated as the bi-dual problem, i.e.,*

$$D_{\text{ReLU}}(t) = \min_{\{\mathbf{W}_{1,j}\}_{j=1}^{m^*}, \mathbf{W}_2 \in \mathbb{R}^{m_1 \times m^*}, \mathbf{W}_3 \in \mathbb{R}^{m^* \times K}} \sum_{j=1}^K \|\mathbf{w}_{3,j}^{\text{row}}\|_2,$$
$$\text{s.t. } \sum_{j=1}^{m^*} ((\mathbf{X}\mathbf{W}_{1,j})_+\mathbf{w}_{2,j}^{\text{col}})_+\mathbf{w}_{3,j}^{\text{row}} = \mathbf{Y}, \|\mathbf{W}_{1,j}\|_F \le t, \|\mathbf{w}_{2,j}^{\text{col}}\|_2 \le 1. \tag{25}$$

For the case where the data matrix is with rank 1 and the neural network is with scalar output, there is no duality gap.

**Theorem 3** *For a three-layer scalar-output ReLU network, let $\mathbf{X}$ be a rank-one data matrix. Then, strong duality holds, i.e., $P_{\text{ReLU}}(t) = D_{\text{ReLU}}(t)$. Suppose that $\boldsymbol{\lambda}^*$ is the optimal solution to the dual problem $D_{\text{ReLU}}(t)$, then the optimal weights for each layer can be formulated as*

$$\mathbf{W}_1 = t\text{sign}(|(\boldsymbol{\lambda}^*)^T(\mathbf{c})_+| - |(\boldsymbol{\lambda}^*)^T(-\mathbf{c})_+|)\boldsymbol{\rho}_0\boldsymbol{\rho}_1^T, \mathbf{w}_2 = \boldsymbol{\rho}_1.$$

*Here $\boldsymbol{\rho}_0 = \mathbf{a}_0/\|\mathbf{a}_0\|_2$ and $\boldsymbol{\rho}_1 \in \mathbb{R}_+^{m_l}$ satisfies $\|\boldsymbol{\rho}_1\| = 1$.*

Now, we consider the case where the data matrix is whitened, i.e., $\mathbf{X}\mathbf{X}^T = \mathbf{I}_n$ and $\mathbf{Y}$ has orthogonal columns. We leave the proof of Theorem 4 and the characterization of optimal solutions in Appendix C.3.

**Theorem 4** *Let* $\{\mathbf{X}, \mathbf{Y}\}$ *be a dataset such that* $\mathbf{X}\mathbf{X}^T = \mathbf{I}_n$ *and* $\mathbf{Y}$ *has orthogonal columns. Then, the optimal value of* $P_{\text{ReLU}}(t)$ *and* $D_{\text{ReLU}}(t)$ *are given by*

$$P_{\text{ReLU}}(t) = t^{-1} \left( \sum_{j=1}^{K} \left( \|(\mathbf{y}_j)_+\|_2^{2/3} + \|(-\mathbf{y}_j)_+\|_2^{2/3} \right) \right)^{3/2}, \tag{26}$$

*and*

$$D_{\text{ReLU}}(t) = t^{-1} \sum_{j=1}^{n} \left( \|(\mathbf{y}_j)_+\|_2 + \|(-\mathbf{y}_j)_+\|_2 \right). \tag{27}$$

Suppose that $\mathbf{Y}$ is the one-hot encoding of the label. Then, we note that $\mathbf{y}_j \geq 0$ and $\|(\mathbf{y}_j)_+\|_2$ is the square root of the number of data points in the $j$-th class. Therefore, we recover the result about the duality gap of standard three-layer ReLU networks.

### 4.2 PARALLEL DEEP RELU NETWORKS

For the parallel architecture, we show that there is no duality gap for arbitrary deep ReLU network with large enough number of branches. Consider the following minimal norm problem:

$$P_{\text{ReLU}}^{\text{prl}} = \min \frac{1}{2} \sum_{l=1}^{L-1} \sum_{j=1}^{m} \|\mathbf{W}_{l,j}\|_F^L + \|\mathbf{W}_L\|_F^L,$$

$$\text{s.t.} \ \sum_{j=1}^{m} (((\mathbf{X}\mathbf{W}_{1,j})_+ \dots \mathbf{W}_{L-2,j})_+ \mathbf{w}_{L-1,j}^{\text{col}})_+ \mathbf{w}_{L,j}^{\text{row}} = \mathbf{Y}. \tag{28}$$

**Proposition 6** *The problem* (28) *can be reformulated as*

$$P_{\text{ReLU}}^{\text{prl}} = \min \frac{L}{2} \sum_{j=1}^{m} \|\mathbf{w}_{L,j}^{\text{row}}\|_2,$$

$$s.t. \ \sum_{j=1}^{m} (((\mathbf{X}\mathbf{W}_{1,j})_+ \dots \mathbf{W}_{L-2,j})_+ \mathbf{w}_{L-1,j}^{\text{col}})_+ \mathbf{w}_{L,j}^{\text{row}} = \mathbf{Y},$$

$$\|\mathbf{W}_{l,j}\|_F \leq 1, l \in [L-2], \|\mathbf{w}_{L-1,j}^{\text{col}}\|_2 \leq 1, j \in [m]. \tag{29}$$

*The dual problem* (29) *is a convex problem defined as*

$$D_{\text{ReLU}}^{\text{prl}} = \max \text{tr}(\mathbf{\Lambda}^T \mathbf{Y}), \ s.t. \max_{\substack{v=((\mathbf{X}\mathbf{W}_1)_+ \dots \mathbf{W}_{L-2})_+ \mathbf{w}_{L-1})_+, \\ \|\mathbf{W}_l\|_F \leq 1, l \in [L-2], \|\mathbf{w}_{L-1}\|_2 \leq 1}} \|\mathbf{\Lambda}^T \mathbf{v}\|_2 \leq L/2. \tag{30}$$

For deep ReLU network with parallel architecture, the strong duality holds.

**Theorem 5** *The strong duality holds for* (29) *in the sense that* $P_{\text{ReLU}}^{\text{prl}} = D_{\text{ReLU}}^{\text{prl}}$ *for* $m \geq m^*$, *where* $m^*$ *is upper bounded by* $KN + 1$.

Similar to case of parallel deep linear networks, the parallel deep ReLU network also achieves zero-duality gap. Therefore, to find the global optimum for parallel deep ReLU network is equivalent to solve a convex program.

## 5 CONCLUSION

We present the convex duality framework for standard neural networks, considering both multi-layer linear networks and three-layer ReLU networks with rank-1 or whitened data. In stark contrast to the two-layer case, the duality gap can be non-zero for neural networks with depth three or more. Meanwhile, for neural networks with parallel architecture, with the regularization of $L$-th power of Frobenius norm in the parameters, we show that strong duality holds and the duality gap reduces to zero. A limitation of our work is that we primarily focus on minimal norm interpolation problems. We expect to generalize our results to general regularized training problems.

## 6 ACKNOWLEDGEMENTS

This work was partially supported by the National Science Foundation under grants ECCS-2037304, DMS-2134248, the Army Research Office.

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

# A  STAIRS OF DUALITY GAP FOR STANDARD DEEP LINEAR NETWORKS

Now we consider partially dualizing the non-convex optimization problem by exchanging a subset of the minimization problems with respect to the hidden layers. Consider the Lagrangian for the primal problem of standard deep linear network

$$P_{\text{lin}}(t) = \min_{\{\mathbf{W}_l\}_{l=1}^{L-1}} \max_{\mathbf{\Lambda}} \text{tr}(\mathbf{\Lambda}^T \mathbf{Y}) - \mathbb{I}\left(\|\mathbf{\Lambda}^T \mathbf{X} \mathbf{W}_1 \dots \mathbf{W}_{L-2} \mathbf{w}_{L-1}\|_2 \le 1\right),$$

$$\text{s.t. } \|\mathbf{W}_i\|_F \le t, i \in [L-2], \|\mathbf{w}_{L-1}\|_2 \le 1. \tag{31}$$

By changing the order of $L-2$ mins and the max in (31), for $l = 0, 1, \dots, L-2$, we can define the $l$-th partial "dual" problem

$$D_{\text{lin}}^{(l)}(t) = \min_{\mathbf{W}_1, \dots, \mathbf{W}_l} \max_{\mathbf{\Lambda}} \min_{\mathbf{W}_{l+1}, \dots, \mathbf{W}_{L-2}} \text{tr}(\mathbf{\Lambda}^T \mathbf{Y}) - \mathbb{I}\left(\|\mathbf{\Lambda}^T \mathbf{X} \mathbf{W}_1 \dots \mathbf{W}_{L-2} \mathbf{w}_{L-1}\|_2 \le 1\right),$$

$$\text{s.t. } \|\mathbf{W}_i\|_F \le t, i \in [L-2], \|\mathbf{w}_{L-1}\|_2 \le 1. \tag{32}$$

For $l = 0$, $D_{\text{lin}}^{(l)}(t)$ corresponds the primal problem $P_{\text{lin}}(t)$, while for $l = L-2$, $D_{\text{lin}}^{(l)}(t)$ is the dual problem $D_{\text{lin}}(t)$. From the following proposition, we illustrate that the dual problem of $D_{\text{lin}}^{(l)}(t)$ corresponds to a minimal norm problem of a neural network with parallel structure.

**Proposition 7** *There exists $m^* \le KN + 1$ such that the problem $D_{\text{lin}}^{(l)}(t)$ is equivalent to the "bi-dual" problem*

$$\min \sum_{j=1}^{m^*} \|\mathbf{w}_{L,j}^{\text{row}}\|_2,$$

$$s.t. \sum_{j=1}^{m^*} \mathbf{X} \mathbf{W}_1 \dots \mathbf{W}_l \mathbf{W}_{l+1,j} \dots \mathbf{W}_{L-2,j} \mathbf{w}_{L-1,j}^{\text{col}} \mathbf{w}_{L,j}^{\text{row}} = \mathbf{Y}, \tag{33}$$

$$\|\mathbf{W}_i\|_F \le t, i \in [l], \|\mathbf{W}_{i,j}\|_F \le t, i = l+1, \dots, L-2, j \in [m^*],$$

$$\|\mathbf{w}_{L-1,j}^{\text{col}}\|_2 \le 1, j \in [m^*],$$

*where the variables are $\mathbf{W}_i \in \mathbb{R}^{m_{i-1} \times m_i}$ for $i \in [l]$, $\mathbf{W}_{i,j} \in \mathbb{R}^{m_{i-1} \times m_i}$ for $i = l+1, \dots, L-2$, $j \in [m^*]$, $\mathbf{W}_{L-1} \in \mathbb{R}^{m_{L-2} \times m^*}$ and $\mathbf{W}_L \in \mathbb{R}^{m^* \times m_L}$.*

We can interpret the problem (33) as the minimal norm problem of a linear network with parallel structures in $(l+1)$-th to $(L-2)$-th layers. This indicates that for $l = 0, 1, \dots, L-2$, the bi-dual formulation of $D_{\text{lin}}^{(l)}(t)$ can be viewed as an interpolation from a network with standard structure to a network with parallel structure. Now, we calculate the exact value of $D_{\text{lin}}^{(l)}(t)$.

**Proposition 8** *The optimal value $D_{\text{lin}}^{(l)}(t)$ follows*

$$D_{\text{lin}}^{(l)}(t) = t^{-(L-2)} \|\mathbf{X}^\dagger \mathbf{Y}\|_{S_{2/(l+2)}}. \tag{34}$$

Suppose that the eigenvalues $\mathbf{X}^\dagger \mathbf{Y}$ are not identical to each other. Then, we have

$$P_{\text{lin}}(t) = D_{\text{lin}}^{(L-2)}(t) > D_{\text{lin}}^{(L-3)}(t) > \dots > D_{\text{lin}}^{(0)}(t) = D(t). \tag{35}$$

In Figure 3, we plot $D_{\text{lin}}^{(l)}(t)$ for $l = 0, \dots, 5$ for an example.

# B  PROOFS OF MAIN RESULTS FOR LINEAR NETWORKS

## B.1  PROOF OF PROPOSITION 1

Consider the Lagrangian function

$$L(\mathbf{W}_1, \dots, \mathbf{W}_L, \mathbf{\Lambda}) = \sum_{j=1}^K \|\mathbf{w}_{L,j}\|_2 + \text{tr}(\mathbf{\Lambda}^T(\mathbf{Y} - \mathbf{X} \mathbf{W}_1 \dots \mathbf{W}_L)). \tag{36}$$

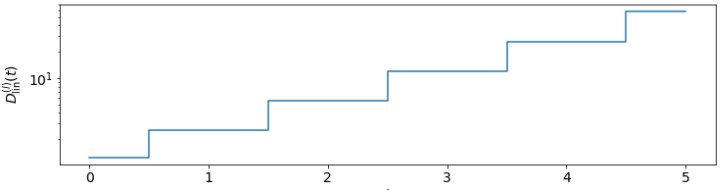

Figure 3: Example of $D_{\text{lin}}^{(l)}(t)$.

Here $\Lambda \in \mathbb{R}^{N \times K}$ is the dual variable. We note that

$$
\begin{aligned}
P(t) = \min_{\mathbf{W}_1, \ldots, \mathbf{W}_L} \; &\max_{\Lambda} \; L(\mathbf{W}_1, \ldots, \mathbf{W}_L, \Lambda), \\
&\text{s.t. } \|\mathbf{W}_i\|_F \le t, i \in [L-2], \|\mathbf{w}_{L-1,j}^{\text{col}}\|_2 \le 1, j \in [m_{L-1}], \\
= \min_{\mathbf{W}_1, \ldots, \mathbf{W}_{L-1}} \; &\max_{\Lambda} \; \text{tr}(\Lambda^T \mathbf{Y}) - \sum_{j=1}^{m_{L-1}} \mathbb{I}\left(\|\Lambda^T \mathbf{X} \mathbf{W}_1 \ldots \mathbf{W}_{L-2} \mathbf{w}_{L-1,j}\|_2 \le 1\right), \\
&\text{s.t. } \|\mathbf{W}_i\|_F \le t, i \in [L-2], \|\mathbf{w}_{L-1,j}^{\text{col}}\|_2 \le 1, j \in [m_{L-1}], \\
= \min_{\mathbf{W}_1, \ldots, \mathbf{W}_{L-2}, \mathbf{W}_{L-1}} \; &\max_{\Lambda} \; \text{tr}(\Lambda^T \mathbf{Y}) - \mathbb{I}\left(\|\Lambda^T \mathbf{X} \mathbf{W}_1 \ldots \mathbf{W}_{L-2} \mathbf{w}_{L-1}\|_2 \le 1\right), \\
&\text{s.t. } \|\mathbf{W}_i\|_F \le t, i \in [L-2], \|\mathbf{w}_{L-1}\|_2 \le 1.
\end{aligned}
\tag{37}
$$

Here $\mathbb{I}(A)$ is 0 if the statement $A$ is true. Otherwise it is $+\infty$. For fixed $\mathbf{W}_1, \ldots, \mathbf{W}_{L-1}$, the constraint on $\mathbf{W}_L$ is linear so we can exchange the order of $\max_{\Lambda}$ and $\min_{\mathbf{W}_L}$ in the second line of (37).

By exchanging the order of $\min$ and $\max$, we obtain the dual problem

$$
\begin{aligned}
D(t) = \max_{\Lambda} \min_{\mathbf{W}_1, \ldots, \mathbf{W}_{L-2}} \; &\text{tr}(\Lambda^T \mathbf{Y}) - \mathbb{I}\left(\|\Lambda^T \mathbf{X} \mathbf{W}_1 \ldots \mathbf{W}_{L-2} \mathbf{w}_{L-1}\|_2 \le 1\right), \\
&\text{s.t. } \|\mathbf{W}_i\|_F \le t, i \in [L-2], \|\mathbf{w}_{L-1}\|_2 \le 1, \\
= \max_{\Lambda} \; &\text{tr}(\Lambda^T \mathbf{Y}) \\
&\text{s.t. } \|\Lambda^T \mathbf{X} \mathbf{W}_1 \ldots \mathbf{W}_{L-2} \mathbf{w}_{L-1}\|_2 \le 1 \\
&\forall \|\mathbf{W}_i\|_F \le t, i \in [L-2], \|\mathbf{w}_{L-1}\|_2 \le 1.
\end{aligned}
\tag{38}
$$

Now we derive the bi-dual problem. The dual problem can be reformulated as

$$
\begin{aligned}
\max_{\Lambda} \; &\text{tr}(\Lambda^T \mathbf{Y}), \\
&\text{s.t. } \|\Lambda^T \mathbf{X} \mathbf{W}_1 \ldots \mathbf{W}_{L-2} \mathbf{w}_{L-1}\|_2 \le 1, \\
&\forall (\mathbf{W}_1, \ldots, \mathbf{W}_{L-2}, \mathbf{w}_{L-1}) \in \Theta.
\end{aligned}
\tag{39}
$$

Here the set $\Theta$ is defined as

$$
\Theta = \{(\mathbf{W}_1, \ldots, \mathbf{W}_{L-2}, \mathbf{w}_{L-1}) \,|\, \|\mathbf{W}_i\|_F \le t, i \in [L-2], \|\mathbf{w}_{L-1}\|_2 \le 1\}.
\tag{40}
$$

By writing $\theta = (\mathbf{W}_1, \ldots, \mathbf{W}_{L-2}, \mathbf{w}_{L-1})$, the dual of the problem (39) is given by

$$
\begin{aligned}
&\min \|\boldsymbol{\mu}\|_{\text{TV}}, \\
&\text{s.t. } \int_{\theta \in \Theta} \mathbf{X} \mathbf{W}_1 \ldots \mathbf{W}_{L-2} \mathbf{w}_{L-1} d\boldsymbol{\mu}(\theta) = \mathbf{Y}.
\end{aligned}
\tag{41}
$$

Here $\boldsymbol{\mu} : \Sigma \to \mathbb{R}^K$ is a signed vector measure and $\Sigma$ is a $\sigma$-field of subsets of $\Theta$. The norm $\|\boldsymbol{\mu}\|_{\text{TV}}$ is the total variation of $\boldsymbol{\mu}$, which can be calculated by

$$
\|\boldsymbol{\mu}\|_{TV} = \sup_{u : \|u(\theta)\|_2 \le 1} \left\{ \int_{\Theta} u^T(\theta) d\mu(\theta) =: \sum_{i=1}^{K} \int_{\Theta} u_i(\theta) d\mu_i(\theta) \right\},
\tag{42}
$$

where we write $\boldsymbol{\mu} = \begin{bmatrix} \mu_1 \\ \vdots \\ \mu_K \end{bmatrix}$. The formulation in (41) has infinite width in each layer. According to Theorem 9 in Appendix E, the measure $\boldsymbol{\mu}$ in the integral can be represented by finitely many Dirac delta functions. Therefore, we can rewrite the problem (41) as

$$
\begin{aligned}
&\min \sum_{j=1}^{m^*} \|\mathbf{w}_{L,j}^{\mathrm{row}}\|_2, \\
&\text{s.t. } \sum_{j=1}^{m^*} \mathbf{X}\mathbf{W}_{1,j}\ldots\mathbf{W}_{L-2,j}\mathbf{w}_{L-1,j}^{\mathrm{col}}\mathbf{w}_{L,j}^{\mathrm{row}} = \mathbf{Y}, \\
&\|\mathbf{W}_{i,j}\|_F \le t, i \in [L-2], \|\mathbf{w}_{L-1,j}^{\mathrm{col}}\|_2 \le 1, j \in [m^*].
\end{aligned}
\tag{43}
$$

Here the variables are $\mathbf{W}_{i,j}$ for $i \in [L-2]$ and $j \in [m^*]$, $\mathbf{W}_{L-1}$ and $\mathbf{W}_L$. As the strong duality holds for the problem (43) and (39), we can reformulate the problem of $D_{\mathrm{lin}}(t)$ as the bi-dual problem (43).

## B.2   Proof of Proposition 2

We restate Proposition 2 with details.

**Proposition 9** *Suppose that $\mathbf{W} \in \mathbb{R}^{d \times K}$ with rank $r$ is given. Consider the following optimization problem:*

$$
\min \frac{1}{2}\left(\|\mathbf{W}_1\|_F^2 + \cdots + \|\mathbf{W}_L\|_F^2\right), \text{ s.t. } \mathbf{W}_1\mathbf{W}_2\ldots\mathbf{W}_L = \mathbf{W},
\tag{44}
$$

*in variables $\mathbf{W}_i \in \mathbb{R}^{m_{i-1} \times m_i}$. Here $m_0 = d$, $m_L = K$ and $m_i \ge r$ for $i = 1, \ldots, L-1$. Then, the optimal value of the problem (44) is given by*

$$
\frac{L}{2}\|\mathbf{W}\|_{S_{2/L}}^{2/L}.
\tag{45}
$$

*Suppose that $\mathbf{W} = \mathbf{U}\boldsymbol{\Sigma}\mathbf{V}^T$. The optimal value can be achieved when*

$$
\mathbf{W}_i = \mathbf{U}_{i-1}\boldsymbol{\Sigma}^{1/L}\mathbf{U}_i^T, \quad i = 1, \ldots, N, \mathbf{U}_0 = \mathbf{U}, \mathbf{U}_L = \mathbf{V}.
\tag{46}
$$

*Here $\mathbf{U}_i \in \mathbb{R}^{r \times m_i}$ satisfies that $\mathbf{U}_i^T\mathbf{U}_i = I$.*

We start with two lemmas.

**Lemma 1** *Suppose that $A \in \mathbb{S}^{n \times n}$ is a positive semi-definite matrix. Then, for any $0 < p < 1$, we have*

$$
\sum_{i=1}^n A_{ii}^p \ge \sum_{i=1}^n \lambda_i(A)^p.
\tag{47}
$$

*Here $\lambda_i$ is the $i$-th largest eigenvalue of $A$.*

**Lemma 2** *Suppose that $P \in \mathbb{R}^{d \times d}$ is a projection matrix. Then, for arbitrary $W \in \mathbb{R}^{d \times K}$, we have*

$$
\sigma_i(PW) \le \sigma_i(W),
$$

*where $\sigma_i(W)$ represents the $i$-th largest singular value of $W$.*

Now, we present the proof for Proposition 2. For $L = 1$, the statement apparently holds. Suppose that for $L = l$ this statement holds. For $L = l + 1$, by writing $\mathbf{A} = \mathbf{W}_2\ldots\mathbf{W}_{l+1}$, we have

$$
\begin{aligned}
&\min \|\mathbf{W}_1\|_F^2 + \cdots + \|\mathbf{W}_L\|_F^2, \text{ s.t. } \mathbf{W}_1\mathbf{W}_2\ldots\mathbf{W}_{l+1} = \mathbf{W} \\
&= \min \|\mathbf{W}_1\|_F^2 + l\|\mathbf{A}\|_{2/l}^{2/l}, \text{ s.t. } \mathbf{W}_1\mathbf{A} = \mathbf{W}, \\
&= \min t^2 + l\|\mathbf{A}\|_{2/l}^{2/l}, \text{ s.t. } \mathbf{W}_1\mathbf{A} = \mathbf{W}, \|\mathbf{W}_1\|_F \le t.
\end{aligned}
\tag{48}
$$

Suppose that $t$ is fixed. It is sufficient to consider the following problem:

$$
\min \|\mathbf{A}\|_{2/l}^{2/l}, \text{ s.t. } \mathbf{W}_1\mathbf{A} = \mathbf{W}, \|\mathbf{W}_1\|_F \le t.
\tag{49}
$$

Suppose that there exists $\mathbf{W}_1$ and $\mathbf{A}$ such that $\mathbf{W} = \mathbf{W}_1\mathbf{A}$. Then, we have $\mathbf{W}\mathbf{A}^\dagger\mathbf{A} = \mathbf{W}_1\mathbf{A}\mathbf{A}^\dagger\mathbf{A} = \mathbf{W}$. As $\mathbf{W}\mathbf{A}^\dagger = \mathbf{W}_1\mathbf{A}\mathbf{A}^\dagger$, according to Lemma 2, $\|\mathbf{W}\mathbf{A}^\dagger\|_F \leq \|\mathbf{W}_1\|_F \leq t$. Therefore, $(\mathbf{W}\mathbf{A}^\dagger, \mathbf{A})$ is also feasible for the problem (49). Hence, the problem (49) is equivalent to

$$\min \|\mathbf{A}\|_{2/l}^{2/l}, \text{ s.t. } \mathbf{W}\mathbf{A}^\dagger\mathbf{A} = \mathbf{W}, \|\mathbf{W}\mathbf{A}^\dagger\|_F \leq t. \tag{50}$$

Assume that $\mathbf{W}$ is with rank $r$. Suppose that $\mathbf{A} = \mathbf{U}\mathbf{\Sigma}\mathbf{V}^T$, where $\mathbf{\Sigma} \in \mathbb{R}^{r_0 \times r_0}$. Here $r_0 \geq r$. Then, we have $\mathbf{A}^\dagger = \mathbf{V}\mathbf{\Sigma}^{-1}\mathbf{U}^T$. We note that

$$\begin{aligned}
&\|\mathbf{W}\mathbf{A}^\dagger\|_F^2 \\
&= \text{tr}(\mathbf{W}\mathbf{V}\mathbf{\Sigma}^{-2}\mathbf{V}^T\mathbf{W}^T) \\
&= \text{tr}(\mathbf{V}^T\mathbf{W}^T\mathbf{W}\mathbf{V}\mathbf{\Sigma}^{-2})
\end{aligned} \tag{51}$$

Denote $G(\mathbf{V}) = \mathbf{V}^T\mathbf{W}^T\mathbf{W}\mathbf{V}$. This implies that

$$\sum_{i=1}^{r} \sigma_i(\mathbf{A})^{-2}\left(G(\mathbf{V})\right)_{ii} \leq t^2.$$

Therefore, we have

$$\left(\sum_{i=1}^{r_0} \sigma_i(\mathbf{A})^{-2}\left(G(\mathbf{V})\right)_{ii}\right)\left(\sum_{i=1}^{r_0} \sigma_i(\mathbf{A})^{2/l}\right)^l \geq \left(\sum_{i=1}^{r_0} (G(\mathbf{V}))_{ii}^{1/(l+1)}\right)^{l+1}.$$

As $\mathbf{W}\mathbf{V}^T\mathbf{V} = \mathbf{W}$, the non-zero eigenvalues of $G(\mathbf{V})$ are exactly the non-zero eigenvalues of $\mathbf{W}\mathbf{V}\mathbf{V}^T\mathbf{W}^T = \mathbf{W}\mathbf{W}^T$, i.e., the square of non-zero singular values of $\mathbf{W}$. From Lemma 1, we have

$$\sum_{i=1}^{r_0} (G(\mathbf{V}))_{ii}^{1/(l+1)} \geq \sum_{i=1}^{r_0} \lambda_i(G(\mathbf{V}))^{1/(l+1)} \geq \sum_{i=1}^{r} \sigma_i(\mathbf{W})^{2/(l+1)}. \tag{52}$$

Therefore, we have

$$\|\mathbf{A}\|_{S_{2/l}}^{2/l} = \sum_{i=1}^{r_0} \sigma_i(\mathbf{A})^{2/l} \geq t^{-2/l}\left(\sum_{i=1}^{r} \sigma_i(\mathbf{W})^{2/(l+1)}\right)^{(l+1)/l} \tag{53}$$

This also implies that

$$\begin{aligned}
&\min \|\mathbf{A}\|_{2/l}^{2/l}, \text{ s.t. } \mathbf{W}_1\mathbf{A} = \mathbf{W}, \|\mathbf{W}_1\|_F \leq t \\
&\geq t^{-2/l}\left(\sum_{i=1}^{r} \sigma_i(\mathbf{W})^{2/(l+1)}\right)^{(l+1)/l}.
\end{aligned} \tag{54}$$

Suppose that $\mathbf{W} = \sum_{i=1}^{r} u_i\sigma_i v_i^T$ is the SVD of $\mathbf{W}$. We can let

$$\begin{aligned}
\mathbf{A} &= \frac{\left(\sum_{i=1}^{r} \sigma_i^{2/(l+1)}\right)^{1/2}}{t}\sum_{i=1}^{r} u_i\sigma_i^{l/(l+1)}\rho_i^T, \\
\mathbf{W}_1 &= \frac{t}{\left(\sum_{i=1}^{r} \sigma_i^{2/(l+1)}\right)^{1/2}}\sum_{i=1}^{r} \rho_i\sigma_i^{1/(l+1)}v_i^T.
\end{aligned} \tag{55}$$

Here $\|\rho_i\|_2 = 1$ and $\rho_i^T\rho_j = 0$ for $i \neq j$. Then, $\mathbf{W}_1\mathbf{A} = \mathbf{W}$ and $\|\mathbf{W}_1\|_F \leq t$. We also have

$$\begin{aligned}
\|\mathbf{A}\|_{S_{2/L}}^{2/L} &= t^{-2/l}\left(\sum_{i=1}^{r} \sigma_i^{2/(l+1)}\right)^{1/l}\sum_{i=1}^{r} \sigma_i^{2/(l+1)} \\
&= t^{-2/l}\left(\sum_{i=1}^{r} \sigma_i(\mathbf{W})^{2/(l+1)}\right)^{(l+1)/l}.
\end{aligned}$$

In summary, we have

$$\min t^2 + l\|\mathbf{A}\|_{2/l}^{S_{2/l}}, \text{ s.t. } \mathbf{W}_1\mathbf{A} = \mathbf{W}, \|\mathbf{W}_1\|_F \leq t.$$

$$= \min_{t>0} t^2 + lt^{-2/l}\left(\sum_{i=1}^{r} \sigma_i(\mathbf{W})^{2/(l+1)}\right)^{(l+1)/l}$$

$$= (l+1)\left(\sum_{i=1}^{r} \sigma_i(\mathbf{W})^{2/(l+1)}\right)^{(l+1)/2} \tag{56}$$

$$= \|\mathbf{W}\|_{S_{2/(l+1)}}^{2/(l+1)}.$$

This completes the proof.

### B.3   PROOF OF THEOREM 1

We first compute the optimal value $P$ of the primal problem. From Proposition 2, the minimal norm problem (7) is equivalent to

$$\min L\|\mathbf{W}\|_{S_{2/L}}^{2/L}, \text{ s.t. } \mathbf{XW} = \mathbf{Y}, \tag{57}$$

in variable $\mathbf{W} \in \mathbb{R}^{d \times K}$. According to Lemma 2, for any feasible $\mathbf{W}$ satisfying $\mathbf{XW} = \mathbf{Y}$, because $\mathbf{X}^\dagger\mathbf{XW} = \mathbf{X}^\dagger\mathbf{Y}$ and $\mathbf{X}^\dagger\mathbf{X}$ is a projection matrix, we have

$$L\|\mathbf{W}\|_{S_{2/L}}^{2/L} \geq L\|\mathbf{X}^\dagger\mathbf{Y}\|_{S_{2/L}}^{2/L}. \tag{58}$$

We also note that $\mathbf{XX}^\dagger\mathbf{Y} = \mathbf{XX}^\dagger\mathbf{XW} = \mathbf{XW} = \mathbf{Y}$. Therefore, $\mathbf{X}^\dagger\mathbf{Y}$ is also feasible for the problem (57). This indicates that $P_{\text{lin}} = \frac{L}{2}\|\mathbf{X}^\dagger\mathbf{Y}\|_{S_{2/L}}^{2/L}$

On the other hand, for a feasible point $(\mathbf{W}_1, \ldots, \mathbf{W}_L)$ for $P_{\text{lin}}(t)$, we note that $(\mathbf{W}_1/t, \ldots, \mathbf{W}_{L-2}/t, \mathbf{W}_{L-1}, t^{L-2}\mathbf{W}_L)$ is feasible for $P_{\text{lin}}(1)$. This implies that $t^{L-2}P_{\text{lin}}(t) = P_{\text{lin}}(1)$, or equivalently, $P_{\text{lin}}(t) = t^{-(L-2)}P_{\text{lin}}(1)$. Recall that

$$P_{\text{lin}} = \min_{t>0} \frac{L-2}{2}t^2 + t^{-(L-2)}P_{\text{lin}}(1)$$

$$= \frac{L}{2}(P_{\text{lin}}(1))^{2/L}. \tag{59}$$

This implies that $P_{\text{lin}}(1) = \|\mathbf{X}^\dagger\mathbf{Y}\|_{S_{2/L}}$ and

$$P_{\text{lin}}(t) = t^{-(L-2)}\|\mathbf{X}^\dagger\mathbf{Y}\|_{S_{2/L}}. \tag{60}$$

For the dual problem $D_{\text{lin}}(t)$ defined in (38), we note that

$$\|\mathbf{\Lambda}^T\mathbf{X}\mathbf{W}_1\ldots\mathbf{W}_{L-2}\mathbf{w}_{L-1}\|_2$$

$$\leq \|\mathbf{\Lambda}^T\mathbf{X}\mathbf{W}_1\ldots\mathbf{W}_{L-2}\|_2\|\mathbf{w}_{L-1}\|_2$$

$$\leq \|\mathbf{\Lambda}^T\mathbf{X}\|_2 \prod_{l=1}^{L-2} \|\mathbf{W}_l\|_2\|\mathbf{w}_{L-1}\|_2 \tag{61}$$

$$\leq \|\mathbf{\Lambda}^T\mathbf{X}\|_2 \prod_{l=1}^{L-2} \|\mathbf{W}_l\|_F\|\mathbf{w}_{L-1}\|_2 = t^{L-2}\|\mathbf{\Lambda}^T\mathbf{X}\|_2.$$

The equality can be achieved when $\mathbf{W}_l = t\mathbf{u}_l\mathbf{u}_{l+1}^T$ for $l \in [L-2]$, where $\|\mathbf{u}_l\|_2 = 1$ for $l = 1, \ldots, L-1$. Specifically, we set $\mathbf{u}_{L-1} = \mathbf{w}_{L-1}$ and let $\mathbf{u}_0$ as right singular vector corresponds to the largest singular value of $\mathbf{\Lambda}^T\mathbf{X}$. Therefore, the constraints on $\mathbf{\Lambda}$ is equivalent to

$$\|\mathbf{\Lambda}^T\mathbf{X}\|_2 \leq t^{-(L-2)}. \tag{62}$$

Thus, according to the Von Neumann's trace inequality, it follows

$$\text{tr}(\mathbf{\Lambda}^T\mathbf{Y}) = \text{tr}(\boldsymbol{\lambda}^T\mathbf{XX}^\dagger\mathbf{Y}) \leq \|\boldsymbol{\lambda}^T\mathbf{X}\|_2\|\mathbf{X}^\dagger\mathbf{Y}\|_* \leq t^{-(L-2)}\|\mathbf{X}^\dagger\mathbf{Y}\|_*. \tag{63}$$

As a result, we have $D_{\text{lin}}(t) = t^{-(L-2)}\|\mathbf{X}^\dagger\mathbf{Y}\|_* < t^{-(L-2)}\|\mathbf{X}^\dagger\mathbf{Y}\|_{S_{2/L}} = P_{\text{lin}}(t)$. Namely, the duality gap exists for the standard neural network.

### B.4 PROOF OF PROPOSITION 3

For simplicity, we write $\mathbf{W}_{L-1,j} = \mathbf{w}_{L-1,j}^{\mathrm{col}}$ and $\mathbf{W}_{L,j} = \mathbf{w}_{L,j}^{\mathrm{row}}$ for $j \in [m]$. For the $j$-th branch of the parallel network, let $\hat{\mathbf{W}}_{l,j} = \alpha_{l,j} \mathbf{W}_{l,j}$ for $l \in [L]$. Here $\alpha_{l,j} > 0$ for $l \in [L]$ and they satisfies that $\prod_{l=1}^{L} \alpha_{l,j} = 1$ for $j \in [m]$. Therefore, we have

$$\mathbf{X}\mathbf{W}_{1,j}\ldots\mathbf{W}_{L-2,j}\mathbf{w}_{L-1,j}^{\mathrm{col}}\mathbf{w}_{L,j}^{\mathrm{row}} = \mathbf{X}\hat{\mathbf{W}}_{1,j}\ldots\hat{\mathbf{W}}_{L-2,j}\hat{\mathbf{w}}_{L-1,j}^{\mathrm{col}}\hat{\mathbf{w}}_{L,j}^{\mathrm{row}}. \tag{64}$$

This implies that $\{\hat{\mathbf{W}}_{l,j}\}_{l\in[L],j\in[m]}$ is also feasible for the problem (14). According to the the inequality of arithmetic and geometric means, the objective function in (14) is lower bounded by

$$\begin{aligned}
&\frac{1}{2}\sum_{j=1}^{m}\sum_{l=1}^{L}\alpha_{l,j}^2\|\mathbf{W}_{l,j}\|_F^2 \\
&\geq \sum_{j=1}^{m}\frac{L}{2}\prod_{l=1}^{L}\left(\alpha_{l,j}^{2/L}\|\mathbf{W}_{l,j}\|_F^{2/L}\right) \\
&= \frac{L}{2}\sum_{j=1}^{m}\prod_{l=1}^{L}\|\mathbf{W}_{l,j}\|_F^{2/L}.
\end{aligned} \tag{65}$$

The equality is achieved when $\alpha_{l,j} = \frac{\prod_{l=1}^{L}\|\mathbf{W}_{l,j}\|_F^{1/L}}{\|\mathbf{W}_{l,j}\|_F}$ for $l \in [L]$ and $j \in [m]$. As the scaling operation does not change $\prod_{l=1}^{L}\|\mathbf{W}_{l,j}\|_F^{2/L}$, we can simply let $\|\mathbf{W}_{l,j}\|_F = 1$ and the lower bound becomes $\frac{L}{2}\sum_{i=1}^{m}\|\mathbf{W}_{L,j}\|_F^{2/L} = \frac{L}{2}\sum_{i=1}^{m}\|\mathbf{w}_{L,j}^{\mathrm{row}}\|_2^{2/L}$. This completes the proof.

### B.5 PROOF OF PROPOSITION 4

We first show that the problem (16) is equivalent to (17). The proof is analogous to the proof of Proposition 3. For simplicity, we write $\mathbf{W}_{L-1,j} = \mathbf{w}_{L-1,j}^{\mathrm{col}}$ and $\mathbf{W}_{L,j} = \mathbf{w}_{L,j}^{\mathrm{row}}$ for $j \in [m]$. Let $\alpha_{l,j} > 0$ for $l \in [L]$ and they satisfies that $\prod_{l=1}^{L} \alpha_{l,j} = 1$ for $j \in [m]$. Consider another parallel network $\{\hat{\mathbf{W}}_{l,j}\}_{l\in[L],j\in[m]}$ whose $j$-th branch is defined by $\hat{\mathbf{W}}_{l,j} = \alpha_{l,j}\mathbf{W}_{l,j}$ for $l \in [L]$. As $\prod_{l=1}^{L} \alpha_{l,j} = 1$, we have

$$\mathbf{X}\mathbf{W}_{1,j}\ldots\mathbf{W}_{L-2,j}\mathbf{w}_{L-1,j}^{\mathrm{col}}\mathbf{w}_{L,j}^{\mathrm{row}} = \mathbf{X}\hat{\mathbf{W}}_{1,j}\ldots\hat{\mathbf{W}}_{L-2,j}\hat{\mathbf{w}}_{L-1,j}^{\mathrm{col}}\hat{\mathbf{w}}_{L,j}^{\mathrm{row}}. \tag{66}$$

This implies that $\{\hat{\mathbf{W}}_{l,j}\}_{l\in[L],j\in[m]}$ is also feasible for the problem (16). According to the the inequality of arithmetic and geometric means, the objective function in (14) is lower bounded by

$$\begin{aligned}
&\frac{1}{2}\sum_{j=1}^{m}\sum_{l=1}^{L}\alpha_{l,j}^L\|\mathbf{W}_{l,j}\|_F^L \\
&\geq \sum_{j=1}^{m}\frac{L}{2}\prod_{l=1}^{L}\left(\alpha_{l,j}\|\mathbf{W}_{l,j}\|_F\right) \\
&= \frac{L}{2}\sum_{j=1}^{m}\prod_{l=1}^{L}\|\mathbf{W}_{l,j}\|_F.
\end{aligned} \tag{67}$$

The equality is achieved when $\alpha_{l,j} = \frac{\prod_{l=1}^{L}\|\mathbf{W}_{l,j}\|_F^{1/L}}{\|\mathbf{W}_{l,j}\|_F}$ for $l \in [L]$ and $j \in [m]$. As the scaling operation does not change $\prod_{l=1}^{L}\|\mathbf{W}_{l,j}\|_F$, we can simply let $\|\mathbf{W}_{l,j}\|_F = 1$ and the lower bound becomes $\frac{L}{2}\sum_{i=1}^{m}\|\mathbf{W}_{L,j}\|_F = \frac{L}{2}\sum_{i=1}^{m}\|\mathbf{w}_{L,j}^{\mathrm{row}}\|_2$. Hence, the problem (16) is equivalent to (17).

For the problem (17), we consider the Lagrangian function

$$L(\mathbf{W}_1,\ldots,\mathbf{W}_L) = \frac{L}{2}\sum_{j=1}^{m}\|\mathbf{w}_{L,j}^{\mathrm{row}}\|_2 + \mathrm{tr}\left(\mathbf{\Lambda}^T(\mathbf{Y} - \sum_{j=1}^{m}\mathbf{X}\mathbf{W}_{1,j}\ldots\mathbf{W}_{L-1,j}^{\mathrm{col}}\mathbf{W}_{L,j}^{\mathrm{row}})\right). \tag{68}$$

The primal problem is equivalent to

$$
\begin{aligned}
P_{\text{lin}}^{\text{prl}} = \min_{\mathbf{W}_1,\dots,\mathbf{W}_L} & \max_{\mathbf{\Lambda}} L(\mathbf{W}_1,\dots,\mathbf{W}_L,\mathbf{\Lambda}), \\
& \text{s.t. } \|\mathbf{W}_{l,j}\|_F \le t, j \in [m_l], l \in [L-2], \|\mathbf{w}_{L-1,j}^{\text{col}}\|_2 \le 1, j \in [m_{L-1}], \\
= \min_{\mathbf{W}_1,\dots,\mathbf{W}_{L-1}} & \max_{\mathbf{\Lambda}} \min_{\mathbf{W}_L} L(\mathbf{W}_1,\dots,\mathbf{W}_L,\mathbf{\Lambda}), \\
& \text{s.t. } \|\mathbf{W}_{l,j}\|_F \le 1, l \in [L-2], \|\mathbf{w}_{L-1,j}^{\text{col}}\|_2 \le 1, j \in [m], \\
= \min_{\mathbf{W}_1,\dots,\mathbf{W}_{L-1}} & \max_{\mathbf{\Lambda}} \operatorname{tr}(\mathbf{\Lambda}^T \mathbf{Y}) - \sum_{j=1}^{m_L} \mathbb{I}\left( \|\mathbf{\Lambda}^T \mathbf{X} \mathbf{W}_{1,j} \dots \mathbf{W}_{L-2,j} \mathbf{w}_{L-1,j}^{\text{col}}\|_2 \le L/2 \right), \\
& \text{s.t. } \|\mathbf{W}_{l,j}\|_F \le 1, l \in [L-2], \|\mathbf{w}_{L-1,j}^{\text{col}}\|_2 \le 1, j \in [m].
\end{aligned}
\tag{69}
$$

The dual problem follows

$$
\begin{aligned}
D_{\text{lin}}^{\text{prl}} = \max_{\mathbf{\Lambda}} & \operatorname{tr}(\mathbf{\Lambda}^T \mathbf{Y}), \\
& \text{s.t. } \|\mathbf{\Lambda}^T \mathbf{X} \mathbf{W}_{1,j} \dots \mathbf{W}_{L-2,j}\|_2 \le L/2, \\
& \quad \forall \|\mathbf{W}_{l,j}\|_F \le 1, l \in [L-2], \|\mathbf{W}_{L-1,j}^{\text{col}}\|_2 \le 1, j \in [m] \\
= \max_{\mathbf{\Lambda}} & \operatorname{tr}(\mathbf{\Lambda}^T \mathbf{Y}), \\
& \text{s.t. } \|\mathbf{\Lambda}^T \mathbf{X} \mathbf{W}_1 \dots \mathbf{W}_{L-2} \mathbf{w}_{L-1}\|_2 \le L/2, \\
& \quad \forall \|\mathbf{W}_i\|_F \le 1, i \in [L-2], \|\mathbf{w}_{L-1}\|_2 \le 1.
\end{aligned}
\tag{70}
$$

### B.6 PROOF OF THEOREM 2

We can rewrite the dual problem as

$$
\begin{aligned}
D_{\text{lin}}^{\text{prl}} = \max_{\mathbf{\Lambda}} & \operatorname{tr}(\mathbf{\Lambda}^T \mathbf{Y}), \\
& \text{s.t. } \|\mathbf{\Lambda}^T \mathbf{X} \mathbf{W}_1 \dots \mathbf{W}_{L-2} \mathbf{w}_{L-1}\|_2 \le L/2, \\
& \quad \forall (\mathbf{W}_1,\dots,\mathbf{W}_{L-2},\mathbf{w}_{L-1}) \in \Theta,
\end{aligned}
\tag{71}
$$

where the set $\Theta$ is defined as

$$
\Theta = \{(\mathbf{W}_1,\dots,\mathbf{W}_{L-2},\mathbf{w}_{L-1}) | \|\mathbf{W}_l\|_F \le 1, l \in [L-2], \|\mathbf{w}_{L-1}\|_2 \le 1\}.
\tag{72}
$$

By writing $\theta = (\mathbf{W}_1,\dots,\mathbf{W}_{L-2},\mathbf{w}_{L-1})$, the bi-dual problem, i.e., the dual problem of (71), is given by

$$
\begin{aligned}
& \min \|\boldsymbol{\mu}\|_{\text{TV}}, \\
& \text{s.t. } \int_{\theta \in \Theta} \mathbf{X} \mathbf{W}_1 \dots \mathbf{W}_{L-2} \mathbf{w}_{L-1} d\boldsymbol{\mu}(\theta) = \mathbf{Y}.
\end{aligned}
\tag{73}
$$

Here $\boldsymbol{\mu} : \Sigma \to \mathbb{R}^K$ is a signed vector measure, where $\Sigma$ is a $\sigma$-field of subsets of $\Theta$ and $\|\boldsymbol{\mu}\|_{\text{TV}}$ is its total variation. The formulation in (73) has infinite width in each layer. According to Theorem 9 in Appendix E, the measure $\boldsymbol{\mu}$ in the integral can be represented by finitely many Dirac delta functions. Therefore, there exists $m^* < KN + 1$ such that we can rewrite the problem (73) as

$$
\begin{aligned}
& \min \sum_{j=1}^{m^*} \|\mathbf{w}_{L,j}^{\text{row}}\|_2, \\
& \text{s.t. } \sum_{j=1}^{m^*} \mathbf{X} \mathbf{W}_{1,j} \dots \mathbf{W}_{L-2,j} \mathbf{w}_{L-1,j}^{\text{col}} \mathbf{w}_{L,j}^{\text{row}} = \mathbf{Y}, \\
& \quad \|\mathbf{W}_{i,j}\|_F \le 1, l \in [L-2], \|\mathbf{w}_{L-1,j}^{\text{col}}\|_2 \le 1, j \in [m^*].
\end{aligned}
\tag{74}
$$

Here the variables are $\mathbf{W}_{l,j}$ for $l \in [L-2]$ and $j \in [m^*]$, $\mathbf{W}_{L-1}$ and $\mathbf{W}_L$. This is equivalent to (17). As the strong duality holds for the problem (71) and (73), the primal problem (17) is equivalent to the dual problem (71) as long as $m \ge m^*$.

Now, we compute the optimal value of $D_{\text{lin}}^{\text{prl}}$. Similar to the proof of Theorem 1, we can show that the constraints in the dual problem (71) is equivalent to

$$\|\mathbf{\Lambda}^T\mathbf{X}\|_2 \leq L/2. \tag{75}$$

Therefore, we have

$$\text{tr}(\mathbf{\Lambda}^T\mathbf{Y}) \leq \|\boldsymbol{\lambda}^T\mathbf{X}\|_2\|\mathbf{X}^\dagger\mathbf{Y}\|_* \leq \frac{L}{2}\|\mathbf{X}^\dagger\mathbf{Y}\|_*. \tag{76}$$

This implies that $P_{\text{lin}}^{\text{prl}} = D_{\text{lin}}^{\text{prl}} = \frac{L}{2}\|\mathbf{X}^\dagger\mathbf{Y}\|_*$.

### B.7 PROOF OF PROPOSITION 8

We note that

$$\begin{aligned}
&\max_{\mathbf{\Lambda}} \text{tr}(\mathbf{\Lambda}^T\mathbf{Y}), \\
&\text{s.t. } \|\mathbf{\Lambda}^T\mathbf{X}\mathbf{W}_1\dots\mathbf{W}_{L-2}\|_2 \leq 1, \|\mathbf{W}_i\|_F \leq t, i = l+1,\dots,L-2 \\
&= \max_{\mathbf{\Lambda}} \min_{\mathbf{W}_{j+1},\dots,\mathbf{W}_{L-2}} \text{tr}(\mathbf{\Lambda}^T\mathbf{Y}), \text{ s.t. } \|\mathbf{\Lambda}^T\mathbf{X}\mathbf{W}_1\dots\mathbf{W}_l\|_2 \leq t^{-(L-2-l)}
\end{aligned} \tag{77}$$

Therefore, we can rewrite $D_{\text{lin}}^{(l)}(t)$ as

$$\begin{aligned}
D_{\text{lin}}^{(l)}(t) = &\min_{\mathbf{W}_1,\dots\mathbf{W}_l} \max_{\mathbf{\Lambda}} \text{tr}(\mathbf{\Lambda}^T\mathbf{Y}), \\
&\text{s.t. } \|\mathbf{\Lambda}^T\mathbf{X}\mathbf{W}_1\dots\mathbf{W}_l\|_2 \leq t^{-(L-2-l)}, \|\mathbf{W}_i\|_F \leq t, i \in [l], \\
= &\min_{\mathbf{W}_1,\dots\mathbf{W}_l} \max_{\mathbf{\Lambda}} t^{-(L-2-l)} \text{tr}(\mathbf{\Lambda}^T\mathbf{Y}), \\
&\text{s.t. } \|\mathbf{\Lambda}^T\mathbf{X}\mathbf{W}_1\dots\mathbf{W}_l\|_2 \leq 1, \|\mathbf{W}_i\|_F \leq t, i \in [l].
\end{aligned} \tag{78}$$

From the equation (11), we note that

$$\begin{aligned}
&\min_{\mathbf{W}_1,\dots\mathbf{W}_j} \max_{\mathbf{\Lambda}} \text{tr}(\mathbf{\Lambda}^T\mathbf{Y}) \\
&\text{s.t. } \|\mathbf{\Lambda}^T\mathbf{X}\mathbf{W}_1\dots\mathbf{W}_j\|_2 \leq 1, \|\mathbf{W}_i\|_F \leq t, i \in [j], \\
&= \min \sum_{j=1}^K \|\mathbf{w}_{l+2,j}\|_2, \\
&\text{s.t. } \|\mathbf{W}_i\|_F \leq t, i \in [L-2], \|\mathbf{w}_{L-1,j}\|_2 \leq 1, j \in [m_{L-1}], \\
&\qquad \mathbf{X}\mathbf{W}_1\dots\mathbf{W}_{l+2} = \mathbf{Y} \\
&= t^{-l}\|\mathbf{X}^\dagger\mathbf{Y}\|_{S_{2/(l+2)}}.
\end{aligned} \tag{79}$$

This completes the proof.

## C PROOFS OF MAIN RESULTS FOR RELU NETWORKS

### C.1 PROOF OF PROPOSITION 5

For the problem of $P(t)$, introduce the Lagrangian function

$$L(\mathbf{W}_1,\mathbf{W}_2,\mathbf{W}_3,\mathbf{\Lambda}) = \sum_{j=1}^K \|\mathbf{w}_{3,j}^{\text{row}}\|_2 - \text{tr}(\mathbf{\Lambda}^T(((\mathbf{X}\mathbf{W}_1)_+\mathbf{W}_2)_+\mathbf{W}_3 - \mathbf{Y})). \tag{80}$$

According to the convex duality of two-layer ReLU network, we have

$$\begin{aligned}
P_{\text{ReLU}}(t) = &\min_{\|\mathbf{W}_1\|_F \leq t, \|\mathbf{w}_2\|\leq 1} \max_{\mathbf{\Lambda}} \text{tr}(\mathbf{\Lambda}^T\mathbf{Y}) - \mathbb{I}(\|\mathbf{\Lambda}^T((\mathbf{X}\mathbf{W}_1)_+\mathbf{w}_2)_+\|_2 \leq 1) \\
= &\min_{\|\mathbf{W}_1\|_F \leq t} \max_{\mathbf{\Lambda}} \min_{\|\mathbf{w}_2\|\leq 1} \text{tr}(\mathbf{\Lambda}^T\mathbf{Y}) - \mathbb{I}(\|\mathbf{\Lambda}^T((\mathbf{X}\mathbf{W}_1)_+\mathbf{w}_2)_+\|_2 \leq 1) \\
= &\min_{\|\mathbf{W}_1\|_F \leq t} \max_{\mathbf{\Lambda}} \text{tr}(\mathbf{\Lambda}^T\mathbf{Y}), \text{ s.t. } \|\mathbf{\Lambda}^T\mathbf{v}\|_2 \leq 1, \forall\mathbf{v} \in \mathcal{A}(\mathbf{W}_1).
\end{aligned} \tag{81}$$

By changing the min and max, we obtain the dual problem.
$$D_{\text{ReLU}}(t) = \max_{\boldsymbol{\Lambda}} \text{tr}(\boldsymbol{\Lambda}^T \mathbf{Y}), \text{ s.t. } \|\boldsymbol{\Lambda}^T \mathbf{v}\|_2 \leq 1, \mathbf{v} \in \mathcal{A}(\mathbf{W}_1), \forall \|\mathbf{W}_1\|_F \leq t. \tag{82}$$

The dual of the dual problem writes
$$\min \|\boldsymbol{\mu}\|_{\text{TV}},$$
$$\text{s.t. } \int_{\|\mathbf{W}_1\|_F \leq t, \|\mathbf{w}_2\|_2 \leq 1} ((\mathbf{X}\mathbf{W}_1)_+ \mathbf{w}_2)_+ \, d\boldsymbol{\mu}(\mathbf{W}_1, \mathbf{w}_2) = \mathbf{Y}. \tag{83}$$

Here $\boldsymbol{\mu}$ is a signed vector measure and $\|\boldsymbol{\mu}\|_{\text{TV}}$ is its total variation. Similar to the proof of Proposition 1, we can find a finite representation for the optimal measure and transform this problem to

$$\min_{\{\mathbf{W}_{1,j}\}_{j=1}^{m^*}, \mathbf{W}_2 \in \mathbb{R}^{m_1 \times m^*}, \mathbf{W}_3 \in \mathbb{R}^{m^* \times K}} \sum_{j=1}^{K} \|\mathbf{w}_{3,j}\|_2,$$

$$\text{s.t. } \sum_{j=1}^{m^*} ((\mathbf{X}\mathbf{W}_{1,j})_+ \mathbf{w}_{2,j})_+ \mathbf{w}_{3,j} = \mathbf{Y}, \|\mathbf{W}_{1,j}\|_F \leq t, \|\mathbf{w}_{2,j}\|_2 \leq 1. \tag{84}$$

Here $m^* \leq KN + 1$. This completes the proof.

## C.2 PROOF OF THEOREM 3

For rank-1 data matrix that $\mathbf{X} = \mathbf{c}\mathbf{a}_0^T$, suppose that $\mathbf{A}_1 = (\mathbf{X}\mathbf{W}_1)_+$. It is easy to observe that
$$\mathbf{A}_1 = (\mathbf{c})_+ \mathbf{a}_{1,+}^T + (-\mathbf{c})_+ \mathbf{a}_{1,-}^T,$$
Here we let $\mathbf{a}_{1,+} = (\mathbf{W}_1 \mathbf{a}_0)_+$ and $\mathbf{a}_{0,-} = (-\mathbf{W}_1 \mathbf{a}_0)_+$.

For a three-layer network, i.e., $L = 3$, suppose that $\boldsymbol{\lambda}^*$ is the optimal solution to the dual problem $D_{\text{ReLU}}(t)$. We consider the extreme points
$$\arg \max_{\|\mathbf{W}_1\|_F \leq t, \|\mathbf{w}_2\|_2 \leq 1} |(\boldsymbol{\lambda}^*)^*((\mathbf{X}\mathbf{W}_1)_+ \mathbf{w}_2)_+|. \tag{85}$$

For fixed $\mathbf{W}_1$, because $\mathbf{a}_{1,+}^T \mathbf{a}_{1,-} = 0$, suppose that
$$\mathbf{w}_2 = u_1 \mathbf{a}_{1,+} + u_2 \mathbf{a}_{1,-} + u_3 \mathbf{r},$$
where $\mathbf{r}^T \mathbf{a}_{1,+} = \mathbf{r}^T \mathbf{a}_{1,-} = 0$ and $\|\mathbf{r}\|_2 = 1$. The maximization problem on $\mathbf{w}_2$ becomes
$$\arg \max \left| (\boldsymbol{\lambda}^*)^T (\mathbf{c})_+ \|\mathbf{a}_{1,+}\|_2^2 (u_1)_+ + (\boldsymbol{\lambda}^*)^T (-\mathbf{c})_+ \|\mathbf{a}_{L-2,-}\|_2^2 (u_2)_+ \right|$$
$$\text{s.t. } u_1^2 \|\mathbf{a}_{1,+}\|_2^2 + u_2^2 \|\mathbf{a}_{1,+}\|_2^2 + u_3^2 \leq 1.$$
If $(\boldsymbol{\lambda}^*)^T (\mathbf{c})_+$ and $(\boldsymbol{\lambda}^*)^T (-\mathbf{c})_+$ have different signs, then the optimal value is
$$\max\{|(\boldsymbol{\lambda}^*)^T (\mathbf{c})_+|\|\mathbf{a}_{L-2,+}\|_2, |(\boldsymbol{\lambda}^*)^T (-\mathbf{c})_+|\|\mathbf{a}_{L-2,-}\|_2\}.$$
And the corresponding optimal $\mathbf{w}_{L-1}$ is $\mathbf{w}_{L-1} = \mathbf{a}_{L-2,+}/\|\mathbf{a}_{L-2,+}\|$ or $\mathbf{w}_{L-1} = \mathbf{a}_{L-2,-}/\|\mathbf{a}_{L-2,-}\|$. Then, the problem becomes
$$\arg \max \max\{|(\boldsymbol{\lambda}^*)^T (\mathbf{c})_+|\|\mathbf{a}_{L-2,+}\|_2, |(\boldsymbol{\lambda}^*)^T (-\mathbf{c})_+|\|\mathbf{a}_{L-2,-}\|_2\}.$$
We note that
$$\max\{\|\mathbf{a}_{L-2,+}\|_2, \|\mathbf{a}_{L-2,-}\|_2\} \leq \|\mathbf{W}_1^T \mathbf{a}_0\|_2 \leq \|\mathbf{W}_1\|_2 \|\mathbf{a}_0\|_2 \leq t^* \|\mathbf{a}_0\|_2.$$
Thus the optimal $\mathbf{W}_1$ follows
$$\mathbf{W}_1 = t^* \text{sign}(|(\boldsymbol{\lambda}^*)^T (\mathbf{c})_+| - |(\boldsymbol{\lambda}^*)^T (-\mathbf{c})_+|) \boldsymbol{\rho}_0 \boldsymbol{\rho}_1^T$$
Here $\boldsymbol{\rho}_0 = \mathbf{a}_0/\|\mathbf{a}_0\|_2$ and $\boldsymbol{\rho}_1 \in \mathbb{R}_+^{m_l}$ satisfies $\|\boldsymbol{\rho}_1\| = 1$. This also gives the optimal $\mathbf{w}_2 = \boldsymbol{\rho}_1$.

On the other hand, if $(\boldsymbol{\lambda}^*)^T (\mathbf{c})_+$ and $(\boldsymbol{\lambda}^*)^T (-\mathbf{c})_+$ have same signs, then, the optimal $\mathbf{w}_2$ follows
$$\mathbf{w}_2 = \frac{|(\boldsymbol{\lambda}^*)^T (\mathbf{c})_+| \mathbf{a}_{L-2,+} + |(\boldsymbol{\lambda}^*)^T (-\mathbf{c})_+| \mathbf{a}_{L-2,-}}{\sqrt{((\boldsymbol{\lambda}^*)^T (\mathbf{c})_+)^2 \|\mathbf{a}_{L-2,+}\|_2^2 + ((\boldsymbol{\lambda}^*)^T (-\mathbf{c})_+)^2 \|\mathbf{a}_{L-2,-}\|_2^2}}.$$
The maximization problem is equivalent to
$$\arg \max ((\boldsymbol{\lambda}^*)^T (\mathbf{c})_+)^2 \|\mathbf{a}_{L-2,+}\|_2^2 + ((\boldsymbol{\lambda}^*)^T (\mathbf{c})_-)^2 \|\mathbf{a}_{L-2,-}\|_2^2$$
By noting that
$$\|\mathbf{a}_{L-2,+}\|_2^2 + \|\mathbf{a}_{L-2,+}\|_2^2 = \|\mathbf{W}_1^T \mathbf{a}_0\|_2^2 \leq \|\mathbf{W}_1\|_2^2 \|\mathbf{a}_0\|_2^2 \leq (t^*)^2 \|\mathbf{a}_0\|_2^2,$$
the optimal $\mathbf{W}_1$ follows
$$\mathbf{W}_1 = t \text{sign}(|(\boldsymbol{\lambda}^*)^T (\mathbf{c})_+| - |(\boldsymbol{\lambda}^*)^T (-\mathbf{c})_+|) \boldsymbol{\rho}_0 \boldsymbol{\rho}_1^T.$$
Here $\boldsymbol{\rho}_0 = \mathbf{a}_0/\|\mathbf{a}_0\|_2$ and $\boldsymbol{\rho}_1 \in \mathbb{R}_+^{m_1}$ satisfies $\|\boldsymbol{\rho}_1\| = 1$. This also gives the optimal $\mathbf{w}_2 = \boldsymbol{\rho}_1$.

## C.3 Proof of Theorem 4

We restate Theorem 4 with details as follows.

**Theorem 6** *Let $\{\mathbf{X}, \mathbf{Y}\}$ be a dataset such that $\mathbf{X}\mathbf{X}^T = \mathbf{I}_n$ and $\mathbf{Y}$ has orthogonal columns. Then, the optimal value of $P_{\mathrm{ReLU}}(t)$ and $D_{\mathrm{ReLU}}(t)$ are given by*

$$P_{\mathrm{ReLU}}(t) = t^{-1} \left( \sum_{j=1}^{K} \left( \|(\mathbf{y}_j)_+\|_2^{2/3} + \|(-\mathbf{y}_j)_+\|_2^{2/3} \right) \right)^{3/2}, \tag{86}$$

*and*

$$D_{\mathrm{ReLU}}(t) = t^{-1} \sum_{j=1}^{n} \left( \|(\mathbf{y}_j)_+\|_2 + \|(-\mathbf{y}_j)_+\|_2 \right). \tag{87}$$

*For the bi-dual formulation of $D_{\mathrm{ReLU}}(t)$ defined in (25), the optimal weight matrices for each layer can be constructed as*

$$\mathbf{W}_{1,r} = \frac{\phi_{0,r}}{\|\phi_{0,r}\|_2} \phi_{1,r}^T, \mathbf{w}_{2,r} = \frac{\phi_{1,r}}{\|\phi_{1,r}\|},$$

*for $r = 1, \ldots, 2K$. Here $(\phi_{0,2j-1}, \phi_{0,2j}) = \left( \mathbf{X}^T(\mathbf{y}_j)_+, \mathbf{X}^T(-\mathbf{y}_j)_+ \right)$, $\|\phi_{1,r}\| = t$, $\phi_{1,r} \in \mathbb{R}_+^{m_1}$.*

*For $P_{\mathrm{ReLU}}(t)$, the optimal weight matrices for each layer write*

$$\mathbf{W}_1 = \sum_{r=1}^{2K} g_r \frac{\phi_{0,r}}{\|\phi_{0,r}\|_2} \phi_{1,r}^T, \mathbf{W}_2 = \left[ \phi_{1,1}, \ldots, \phi_{1,2K} \right].$$

*Here*

$$g_{2j+1} = \frac{\|(\mathbf{y}_j)_+\|_2^{1/3}}{\left( \sum_{j=1}^{K} \left( \|(\mathbf{y}_j)_+\|_2^{2/3} + \|(-\mathbf{y}_j)_+\|_2^{2/3} \right) \right)^{3/2}},$$

$$g_{2j+2} = \frac{\|(-\mathbf{y}_j)_+\|_2^{1/3}}{\left( \sum_{j=1}^{K} \left( \|(\mathbf{y}_j)_+\|_2^{2/3} + \|(-\mathbf{y}_j)_+\|_2^{2/3} \right) \right)^{3/2}}.$$

*We require that $\|\phi_{1,j}\|_2 = 1$ for $j \in [2K]$ and $\phi_{1,i}^T \phi_{1,j} = 0$ for $i \neq j$.*

Since $\mathbf{X}\mathbf{X}^T = I_n$, we can characterize the set as

$$\cup_{\|\mathbf{W}_1\|_F \leq t} \mathcal{A}(\mathbf{W}_1) = \{(\mathbf{z})_+ | \mathbf{z} \in \mathbb{R}^N, \|\mathbf{z}\|_2 \leq t\}. \tag{88}$$

Here $\mathcal{A}(\mathbf{W}_1)$ is defined in (23). For one thing, for $\mathbf{z} \in \mathbb{R}^N$, $\|z\|_2 \leq t$, we can let $\mathbf{W}_1 = \mathbf{X}^T \mathbf{z} \mathbf{v}^T$ and $\mathbf{w}_2 = \mathbf{v}$, where $\mathbf{v} \in \mathbb{R}_+^{m_1}$ with $\|\mathbf{v}\|_2 = 1$. Then, $\|\mathbf{W}_1\|_F \leq t$, $\|\mathbf{w}_2\|_2 \leq 1$ and $(\mathbf{X}\mathbf{W}_1)_+ \mathbf{w}_2)_+ = (\mathbf{z})_+$. For another, for any $\|\mathbf{W}_1\|_F \leq t$ and $\|\mathbf{w}_2\| \leq 1$, we note that

$$\|(\mathbf{X}\mathbf{W}_1)_+ \mathbf{w}_2\|_2 \leq \|(\mathbf{X}\mathbf{W}_1)_+\|_2 \leq \|(\mathbf{X}\mathbf{W}_1)_+\|_F \leq \|\mathbf{X}\mathbf{W}_1\|_F \leq \|\mathbf{W}_1\|_F \leq t.$$

Therefore, $\cup_{\|W\|_F \leq t} \mathcal{A}(W) = \{(\mathbf{z})_+ | \mathbf{z} \in \mathbb{R}^N, \|\mathbf{z}\|_2 \leq t\}$.

Thus, the constraint on $\mathbf{\Lambda}$ in the dual problem (24) is equivalent to say that $t\|(\mathbf{\Lambda}^*)^T(\mathbf{z})_+\|_2 \leq 1$ for all $\mathbf{z} \in \mathbb{R}^N$ satisfying $\|\mathbf{z}\|_2 \leq 1$. As $Y$ has orthogonal columns, we note that

$$\mathrm{tr}((\mathbf{\Lambda}^*)^T \mathbf{Y}) = \sum_{j=1}^{K} (\lambda_j^*)^T ((\mathbf{y}_j)_+ - (-\mathbf{y}_j)_+) \leq t^{-1} \sum_{j=1}^{n} \left( \|(\mathbf{y}_j)_+\|_2 + \|(-\mathbf{y}_j)_+\|_2 \right).$$

Suppose that $(\mathbf{y}_j)_+ \neq 0$ and $(-\mathbf{y}_j)_+ \neq 0$ for all $j = 1, \ldots, K$. Then, the dual problem is minimized by

$$\mathbf{\Lambda}^* = t^{-1} \left[ \frac{(\mathbf{y}_1)_+}{\|(\mathbf{y}_1)_+\|_2} - \frac{(-\mathbf{y}_1)_+}{\|(-\mathbf{y}_1)_+\|_2}, \ldots, \frac{(\mathbf{y}_K)_+}{\|(\mathbf{y}_K)_+\|_2} - \frac{(-\mathbf{y}_K)_+}{\|(-\mathbf{y}_K)_+\|_2} \right].$$

We can also verify that

$$t\|(\mathbf{\Lambda}^*)^T(z)_+\|_2 \leq 1, \forall \|z\|_2 \leq 1. \tag{89}$$

Suppose that $(z)_+ = \sum_{j=1}^K \alpha_{j,1} \frac{(\mathbf{y}_j)_+}{\|(\mathbf{y}_j)_+\|_2} + \alpha_{j,2} \frac{(-\mathbf{y}_j)_+}{\|(-\mathbf{y}_j)_+\|_2} + r$, where $r \in \mathbb{R}_+^N$ is orthogonal to $(\mathbf{y}_j)_+$ and $(-\mathbf{y}_j)_+$. Here $\alpha_{j,1}, \alpha_{j,2} \geq 0$. As $\|(z)_+\| \leq \|z\|_2 \leq 1$, we have

$$\sum_{j=1}^K (\alpha_{j,1}^2 + \alpha_{j,2}^2) \leq 1 - \|r\|_2^2 \leq 1.$$

We note that

$$\|(\mathbf{\Lambda}^*)^T t(z)_+\|_2^2 = \sum_{j=1}^K (\alpha_{j,1} - \alpha_{j,2})^2 \leq \sum_{j=1}^K (\alpha_{j,1}^2 + \alpha_{j,2}^2) \leq 1. \tag{90}$$

Thus, $\mathbf{\Lambda}^*$ satisfies the constraint (89).

We can characterize the optimal layer weight to the bi-dual problem as the extreme points that solves

$$\arg\max_{\|\mathbf{W}_1\|_F \leq t, \|\mathbf{w}_2\|_2 \leq 1} \|(\mathbf{\Lambda}^*)^T((\mathbf{X}\mathbf{W}_1)_+\mathbf{w}_2)_+\|_2. \tag{91}$$

These extreme points correspond to the constraints

$$\left\|(\mathbf{\Lambda}^*)^T \frac{t(\mathbf{y}_j)_+}{\|(\mathbf{y}_j)_+\|_2}\right\|_2 \leq 1, \left\|(\mathbf{\Lambda}^*)^T \frac{t(-\mathbf{y}_j)_+}{\|(-\mathbf{y}_j)_+\|_2}\right\|_2 \leq 1.$$

In other words, the dual problem is equivalent to

$$D_{\mathrm{ReLU}}(t) = \max_{\mathbf{\Lambda}} \mathrm{tr}(\mathbf{\Lambda}^T\mathbf{Y}),$$
$$\text{s.t. } \left\|\mathbf{\Lambda}^T \frac{t^*(\mathbf{y}_j)_+}{\|(\mathbf{y}_j)_+\|_2}\right\|_2 \leq 1, \left\|\mathbf{\Lambda}^T \frac{t^*(-\mathbf{y}_j)_+}{\|(-\mathbf{y}_j)_+\|_2}\right\|_2 \leq 1, 1 \leq j \leq K. \tag{92}$$

Now, we consider an arbitrary matrix $\mathbf{W}_1$ satisfying $\|\mathbf{W}_1\|_F \leq t$. Denote

$$P(\mathbf{W}_1) = \max_{\mathbf{\Lambda}} \mathrm{tr}(\mathbf{\Lambda}^T\mathbf{Y}), \text{ s.t. } \|\mathbf{\Lambda}^T\mathbf{v}\|_2 \leq 1, \forall \mathbf{v} \in \mathcal{A}(\mathbf{W}_1). \tag{93}$$

Suppose that $\mathbf{\Lambda}^*$ is the optimal solution. Then, we have

$$\mathrm{tr}((\mathbf{\Lambda}^*)^T\mathbf{Y}) = \sum_{j=1}^K (\boldsymbol{\lambda}_j^*)^T((\mathbf{y}_j)_+ - (-\mathbf{y}_j)_+)$$
$$= \sum_{j=1}^K (\boldsymbol{\lambda}_j^*)^T \frac{(\mathbf{y}_j)_+}{\|(\mathbf{y}_j)_+\|_2} \|(\mathbf{y}_j)_+\|_2$$
$$- \sum_{j=1}^K (\boldsymbol{\lambda}_j^*)^T \frac{(-\mathbf{y}_j)_+}{\|(-\mathbf{y}_j)_+\|_2} \|(-\mathbf{y}_j)_+\|_2.$$

For a vector $\boldsymbol{\lambda} \in \mathbb{R}^N$, define

$$g(\boldsymbol{\lambda}; \mathbf{W}_1) = \max_{\|\mathbf{w}_2\| \leq 1} |\boldsymbol{\lambda}^T((\mathbf{X}\mathbf{W}_1)_+\mathbf{w}_2)_+|. \tag{94}$$

For a given $\mathbf{u} \in \mathbb{R}_+^N$, we want to estimate

$$\max |\mathbf{u}^T\boldsymbol{\lambda}| \text{ s.t. } g(\boldsymbol{\lambda}; \mathbf{W}_1) \leq 1. \tag{95}$$

The following lemma gives an upper bound.

**Lemma 3** *Suppose that $\mathbf{u} \in \mathbb{R}^N$ satisfying $\|\mathbf{u}\|_2 = 1$. Then, for arbitrary $\boldsymbol{\lambda}$ satisfying $g(\boldsymbol{\lambda}; \mathbf{W}_1) \leq 1$, we have $|\mathbf{u}^T\boldsymbol{\lambda}| \leq 1/g(\mathbf{u}; \mathbf{W}_1)$.*

Denote $\mathbf{z}_{2j+1} = \frac{(\mathbf{y}_j)_+}{\|(\mathbf{y}_j)_+\|_2}$ and $\mathbf{z}_{2j+2} = \frac{(-\mathbf{y}_j)_+}{\|(-\mathbf{y}_j)_+\|_2}$. For simplicity, we write $g(\mathbf{z}; \mathbf{W}_1) = g(\mathbf{z})$. We note that $\boldsymbol{\lambda}_j^*$ satisfies that $g(\boldsymbol{\lambda}_j^*) \leq 1$. According to Lemma 2, we have

$$P^*(\mathbf{W}_1) = \mathrm{tr}((\mathbf{\Lambda})^T\mathbf{Y}) \leq \sum_{j=1}^K \left(\frac{\|(\mathbf{y}_j)_+\|_2}{g(\mathbf{z}_{2j+1})} + \frac{\|(\mathbf{y}_j)_+\|_2}{g(\mathbf{z}_{2j+2})}\right).$$

This serves as an upper bound for $P^*(\mathbf{W}_1)$. We note that for $\mathbf{u} \in \mathbb{R}_+^N$, we have

$$
\begin{aligned}
g(\mathbf{u})^2 &= \max_{\|\mathbf{w}_2\| \leq 1} |\mathbf{u}^T((\mathbf{X}\mathbf{W}_1)_+\mathbf{w}_2)_+|^2 \\
&\leq \max_{\|\mathbf{w}_2\| \leq 1} |\mathbf{u}^T(\mathbf{X}\mathbf{W}_1)_+\mathbf{w}_2|^2 \\
&= \|\mathbf{u}^T((\mathbf{X}\mathbf{W}_1)_+\|_2^2 \\
&\leq \|\mathbf{u}^T\mathbf{X}\mathbf{W}_1\|_2^2 \\
&\leq \operatorname{tr}(\mathbf{u}\mathbf{u}^T\mathbf{X}\mathbf{W}_1\mathbf{W}_1^T\mathbf{X}^T).
\end{aligned}
$$

Then, we have

$$
\sum_{j=1}^{2K} g(\mathbf{z}_j)^2 \leq \operatorname{tr}\left(\left(\sum_{j=1}^{2K}\mathbf{z}_j\mathbf{z}_j^T\right)\mathbf{X}\mathbf{W}_1\mathbf{W}_1^T\mathbf{X}\right) \leq \operatorname{tr}(\mathbf{X}\mathbf{W}_1\mathbf{W}_1^T\mathbf{X}^T) = t^2.
$$

According to the Holder's inequality, we have

$$
\begin{aligned}
&\left(\sum_{j=1}^{K}\left(\|(\mathbf{y}_j)_+\|_2/g(\mathbf{z}_{2j+1}) + \|(\mathbf{y}_j)_+\|_2/g(\mathbf{z}_{2j+2})\right)\right)^{2/3}\left(\sum_{j=1}^{2K}g(\mathbf{z}_j)^2\right)^{1/3} \\
&\geq \sum_{j=1}^{K}\left(\|(\mathbf{y}_j)_+\|_2^{2/3} + \|(-\mathbf{y}_j)_+\|_2^{2/3}\right).
\end{aligned}
$$

Thus, it follows

$$
\begin{aligned}
&\sum_{j=1}^{K}\left(\|(\mathbf{y}_j)_+\|_2/g(\mathbf{z}_{2j+1}) + \|(\mathbf{y}_j)_+\|_2/g(\mathbf{z}_{2j+2})\right) \\
&\geq t^{-1}\left(\sum_{j=1}^{K}\left(\|(\mathbf{y}_j)_+\|_2^{2/3} + \|(-\mathbf{y}_j)_+\|_2^{2/3}\right)\right)^{3/2}.
\end{aligned}
$$

The optimal $g(\mathbf{z}_j)$ follows

$$
g(\mathbf{z}_{2j+1}) = \frac{t\|(\mathbf{y}_j)_+\|_2^{1/3}}{\left(\sum_{j=1}^{K}\left(\|(\mathbf{y}_j)_+\|_2^{2/3} + \|(-\mathbf{y}_j)_+\|_2^{2/3}\right)\right)^{3/2}},
$$

$$
g(\mathbf{z}_{2j+2}) = \frac{t\|(-\mathbf{y}_j)_+\|_2^{1/3}}{\left(\sum_{j=1}^{K}\left(\|(\mathbf{y}_j)_+\|_2^{2/3} + \|(-\mathbf{y}_j)_+\|_2^{2/3}\right)\right)^{3/2}}.
$$

This bound can be achieved for

$$
\mathbf{W}_1 = \sum_{j=1}^{2K} g(\mathbf{z}_j)\mathbf{z}_j\boldsymbol{\phi}_{1,j}^T,
$$

where $\|\boldsymbol{\phi}_{1,j}\|_2 = 1$ for $j \in [2K]$ and $\boldsymbol{\phi}_{1,i}^T\boldsymbol{\phi}_{1,j} = 0$ for $i \neq j$.

### C.4 PROOF OF PROPOSITION 6

Analogous to the proof of Proposition 4, we can reformulate (28) into (29). The rest of the proof is analogous to the proof of Proposition 4. For the problem (29), we consider the Lagrangian function

$$
\begin{aligned}
&L(\mathbf{W}_1, \ldots, \mathbf{W}_L) \\
&= \frac{L}{2}\sum_{j=1}^{m}\|\mathbf{w}_{L,j}^{\text{row}}\|_2 + \operatorname{tr}\left(\mathbf{\Lambda}^T(\mathbf{Y} - \sum_{j=1}^{m}(((\mathbf{X}\mathbf{W}_{1,j})_+ \cdots \cdots \mathbf{W}_{L-2,j})_+\mathbf{w}_{L-1,j}^{\text{col}})_+\mathbf{w}_{L,j}^{\text{row}})\right).
\end{aligned}
\tag{96}
$$

The primal problem is equivalent to

$$
\begin{aligned}
& P_{\mathrm{ReLU}}^{\mathrm{prl}} \\
&= \min_{\mathbf{W}_1,\ldots,\mathbf{W}_L} \max_{\boldsymbol{\Lambda}} L(\mathbf{W}_1,\ldots,\mathbf{W}_L,\boldsymbol{\Lambda}), \\
&\quad \text{s.t. } \|\mathbf{W}_{l,j}\|_F \le t, j \in [m_l], l \in [L-2], \|\mathbf{w}_{L-1,j}^{\mathrm{col}}\|_2 \le 1, j \in [m_{L-1}], \\
&= \min_{\mathbf{W}_1,\ldots,\mathbf{W}_{L-1}} \max_{\boldsymbol{\Lambda}} \min_{\mathbf{W}_L} L(\mathbf{W}_1,\ldots,\mathbf{W}_L,\boldsymbol{\Lambda}), \\
&\quad \text{s.t. } \|\mathbf{W}_{l,j}\|_F \le 1, l \in [L-2], \|\mathbf{w}_{L-1,j}^{\mathrm{col}}\|_2 \le 1, j \in [m], \\
&= \min_{\mathbf{W}_1,\ldots,\mathbf{W}_{L-1}} \max_{\boldsymbol{\Lambda}} \mathrm{tr}(\boldsymbol{\Lambda}^T \mathbf{Y}) - \sum_{j=1}^{m} \mathbb{I}\left( \|\boldsymbol{\Lambda}^T(((\mathbf{X}\mathbf{W}_{1,j})_+ \ldots \mathbf{W}_{L-2,j})_+ \mathbf{w}_{L-1,j}^{\mathrm{col}})_+\|_2 \le L/2\right), \\
&\quad \text{s.t. } \|\mathbf{W}_{l,j}\|_F \le 1, l \in [L-2], \|\mathbf{w}_{L-1,j}^{\mathrm{col}}\|_2 \le 1, j \in [m].
\end{aligned}
\tag{97}
$$

By exchanging the order of $\min$ and $\max$, the dual problem follows

$$
\begin{aligned}
D_{\mathrm{ReLU}}^{\mathrm{prl}} &= \max_{\boldsymbol{\Lambda}} \mathrm{tr}(\boldsymbol{\Lambda}^T \mathbf{Y}), \\
&\quad \text{s.t. } \|\boldsymbol{\Lambda}^T(((\mathbf{X}\mathbf{W}_{1,j})_+ \ldots \mathbf{W}_{L-2,j})_+ \mathbf{w}_{L-1,j}^{\mathrm{col}})_+\|_2 \le L/2, \\
&\qquad \forall \|\mathbf{W}_{l,j}\|_F \le 1, l \in [L-2], \|\mathbf{W}_{L-1,j}^{\mathrm{col}}\|_2 \le 1, j \in [m] \\
&= \max_{\boldsymbol{\Lambda}} \mathrm{tr}(\boldsymbol{\Lambda}^T \mathbf{Y}), \\
&\quad \text{s.t. } \|\boldsymbol{\Lambda}^T(((\mathbf{X}\mathbf{W}_1)_+ \ldots \mathbf{W}_{L-2})_+ \mathbf{w}_{L-1})_+\|_2 \le L/2, \\
&\qquad \forall \|\mathbf{W}_i\|_F \le 1, i \in [L-2], \|\mathbf{w}_{L-1}\|_2 \le 1.
\end{aligned}
\tag{98}
$$

## C.5 PROOF OF THEOREM 5

The proof is analogous to the proof of Theorem 2. We can rewrite the dual problem as

$$
\begin{aligned}
D_{\mathrm{ReLU}}^{\mathrm{prl}} &= \max_{\boldsymbol{\Lambda}} \mathrm{tr}(\boldsymbol{\Lambda}^T \mathbf{Y}), \\
&\quad \text{s.t. } \|\boldsymbol{\Lambda}^T(((\mathbf{X}\mathbf{W}_1)_+ \ldots \mathbf{W}_{L-2})_+ \mathbf{w}_{L-1})_+\|_2 \le L/2, \\
&\qquad \forall (\mathbf{W}_1,\ldots,\mathbf{W}_{L-2},\mathbf{w}_{L-1}) \in \Theta,
\end{aligned}
\tag{99}
$$

where the set $\Theta$ is defined as

$$
\Theta = \{(\mathbf{W}_1,\ldots,\mathbf{W}_{L-2},\mathbf{w}_{L-1}) \| \|\mathbf{W}_l\|_F \le 1, l \in [L-2], \|\mathbf{w}_{L-1}\|_2 \le 1\}. \tag{100}
$$

By writing $\theta = (\mathbf{W}_1,\ldots,\mathbf{W}_{L-2},\mathbf{w}_{L-1})$, the bi-dual problem, i.e., the dual problem of (71), is given by

$$
\begin{aligned}
&\min \|\boldsymbol{\mu}\|_{\mathrm{TV}}, \\
&\text{s.t. } \int_{\theta \in \Theta} (((\mathbf{X}\mathbf{W}_1)_+ \ldots \mathbf{W}_{L-2})_+ \mathbf{w}_{L-1})_+ d\boldsymbol{\mu}(\theta) = \mathbf{Y}.
\end{aligned}
\tag{101}
$$

Here $\boldsymbol{\mu} : \Sigma \to \mathbb{R}^K$ is a signed vector measure, where $\Sigma$ is a $\sigma$-field of subsets of $\Theta$ and $\|\boldsymbol{\mu}\|_{\mathrm{TV}}$ is its total variation. The formulation in (101) has infinite width in each layer. According to Theorem 9 in Appendix E, the measure $\boldsymbol{\mu}$ in the integral can be represented by finitely many Dirac delta functions. Therefore, there exists $m^* \le KN + 1$ such that we can rewrite the problem (101) as

$$
\begin{aligned}
&\min \sum_{j=1}^{m^*} \|\mathbf{w}_{L,j}^{\mathrm{row}}\|_2, \\
&\text{s.t. } \sum_{j=1}^{m^*} (((\mathbf{X}\mathbf{W}_{1,j})_+ \ldots \mathbf{W}_{L-2,j})_+ \mathbf{w}_{L-1,j}^{\mathrm{col}})_+ \mathbf{w}_{L,j}^{\mathrm{row}} = \mathbf{Y}, \\
&\qquad \|\mathbf{W}_{i,j}\|_F \le 1, l \in [L-2], \|\mathbf{w}_{L-1,j}^{\mathrm{col}}\|_2 \le 1, j \in [m^*].
\end{aligned}
\tag{102}
$$

Here the variables are $\mathbf{W}_{l,j}$ for $l \in [L-2]$ and $j \in [m^*]$, $\mathbf{W}_{L-1}$ and $\mathbf{W}_L$. This is equivalent to (29). As the strong duality holds for the problem (99) and (101), the primal problem (29) is equivalent to the dual problem (71) as long as $m \ge m^*$.

# D  PROOFS OF AUXILIARY RESULTS

## D.1  PROOF OF LEMMA 1

Denote $a \in \mathbb{R}^n$ such that $a_i = A_{ii}$ and denote $b \in \mathbb{R}^n$ such that $b_i = \lambda_i(A)$. We can show that $a$ is majorized by $b$, i.e., for $k \in [n-1]$, we have

$$\sum_{i=1}^{k} a_{(i)} \leq \sum_{i=1}^{k} b_{(i)}, \tag{103}$$

and $\sum_{i=1}^{n} a_i = \sum_{i=1}^{n} b_i$. Here $a_{(i)}$ is the $i$-th largest entry in $a$. We first note that

$$\sum_{i=1}^{n} A_{ii} = \text{tr}(A) = \sum_{i=1}^{n} \lambda_i(A).$$

On the other hand, for $k \in [n-1]$, we have

$$\begin{aligned}
\sum_{i=1}^{k} a_{(i)} &= \max_{v \in \mathbb{R}^n, v_i \in \{0,1\}, 1^T v = k} v^T a \\
&= \max_{v \in \mathbb{R}^n, v_i \in \{0,1\}, 1^T v = k} \text{tr}(\mathbf{diag}(v) A \mathbf{diag}(v)) \\
&\leq \max_{V \in \mathbb{R}^{k \times n}, VV^T = I} \text{tr}(V A V^T) \\
&= \sum_{i=1}^{k} \lambda_i(A) = \sum_{i=1}^{k} b_{(i)}.
\end{aligned} \tag{104}$$

Therefore, $a$ is majorized by $b$. As $f(x) = -x^p$ is a convex function, according to the Karamata's inequality, we have

$$\sum_{i=1}^{n} f(a_i) \leq \sum_{i=1}^{n} f(b_i).$$

This completes the proof.

## D.2  PROOF OF LEMMA 2

According to the min-max principle for singular value, we have

$$\sigma_i(W) = \min_{\dim(S) = d-i+1} \max_{x \in S, \|x\|_2 = 1} \|Wx\|_2.$$

As $P$ is a projection matrix, for arbitrary $x \in \mathbb{R}^d$, we have $\|PWx\|_2 \leq \|Wx\|_2$. Therefore, we have

$$\max_{x \in S, \|x\|_2 = 1} \|PWx\|_2 \leq \max_{x \in S, \|x\|_2 = 1} \|Wx\|_2.$$

This completes the proof.

## D.3  PROOF OF LEMMA 3

Suppose that $\boldsymbol{\lambda}(\mathbf{u})$ is optimal solution to

$$\arg\max_{\mathbf{v}} |\mathbf{u}^T \boldsymbol{\lambda}| \text{ s.t. } g(\boldsymbol{\lambda}; \mathbf{W}_1) \leq 1. \tag{105}$$

Denote $n_e$ is the number of extreme points of

$$\arg \max_{\mathbf{v} \in \mathcal{A}(W_1)} |(\boldsymbol{\lambda}(\mathbf{u}))^T \mathbf{v}|.$$

Denote $\mathbf{v}_1, \ldots, \mathbf{v}_{n_e}$ as extreme points of (105). We note that (105) is the dual problem for a scalar output three-layer neural network. By applying the previous duality theorem, we can write $\mathbf{u} = \sum_{i=1}^{n_e} a_i \mathbf{v}_i$, where $\mathbf{v}_i \in \mathcal{A}(\mathbf{W}_1)$. We also have $|\mathbf{u}^T \boldsymbol{\lambda}(\mathbf{u})| = \sum_{i=1}^{n_e} |a_i|$. Besides, we note that

$$1 = \|\mathbf{u}\|_2^2 = \sum_{i=1}^{n_e} a_i \mathbf{u}^T \mathbf{v}_i \leq \sum_{i=1}^{n_e} |a_i| g(\mathbf{u}; \mathbf{W}_1) = |\mathbf{u}^T \boldsymbol{\lambda}(\mathbf{u})| g(\mathbf{u}; \mathbf{W}_1).$$

This completes the proof.

# E  CARATHEODORY'S THEOREM AND FINITE REPRESENTATION

We first review a generalized version of Caratheodory's theorem introduced in (Rosset et al., 2007).

**Theorem 7** *Let $\mu$ be a positive measure supported on a bounded subset $D \subseteq \mathbb{R}^N$. Then, there exists a measure $\nu$ whose support is a finite subset of $D$, $\{z_1, \ldots, z_k\}$, with $k \leq N + 1$ such that*

$$\int_D z d\mu(z) = \sum_{i=1}^{k} z_i d\nu(z_i), \tag{106}$$

*and $\|\mu\|_{TV} = \|\nu\|_{TV}$.*

We can generalize this theorem to signed vector measures.

**Theorem 8** *Let $\mu : \Sigma \to \mathbb{R}^K$ be a signed vector measure supported on a bounded subset $D \subseteq \mathbb{R}^N$. Here $\Sigma$ is a $\sigma$-field of subsets of $D$. Then, there exists a measure $\nu$ whose support is a finite subset of $D$, $\{z_1, \ldots, z_k\}$, with $k \leq KN + 1$ such that*

$$\int_D z d\mu(z) = \sum_{i=1}^{k} z_i d\nu(z_i), \tag{107}$$

*and $\|\nu\|_{TV} = \|\mu\|_{TV}$.*

PROOF  Let $\mu$ be a signed vector measure supported on a bounded subset $D \subseteq \mathbb{R}^N$. Consider the extended set $\tilde{D} = \{zu^T | z \in D, u \in \mathbb{R}^K, \|u\|_2 = 1\}$. Then, $\mu$ corresponds to a scalar-valued measure $\tilde{\mu}$ on the set $\tilde{D}$ and $\|\mu\|_{TV} = \|\tilde{\mu}\|_{TV}$. We note that $\tilde{D}$ is also bounded. Therefore, by applying Theorem 7 to the set $\tilde{D}$ and the measure $\tilde{\mu}$, there exists a measure $\tilde{\nu}$ whose support is a finite subset of $\tilde{D}$, $\{z_1 u_1^T, \ldots, z_k u_k^T\}$, with $k \leq KN + 1$ such that

$$\int_{\tilde{D}} Z d\tilde{\mu}(Z) = \sum_{i=1}^{k} z_i u_i^T d\tilde{\nu}(z_i u_i^T), \tag{108}$$

and $\|\tilde{\mu}\|_{TV} = \|\tilde{\nu}\|_{TV}$. We can define $\nu$ as the signed vector measure whose support is a finite subset $\{z_1, \ldots, z_k\}$ and $d\nu(z_i) = u_i d\tilde(z_i u_i)$. Then, $\|\nu\|_{TV} = \|\tilde{\nu}\|_{TV} = \|\tilde{\mu}\|_{TV} = \|\mu\|_{TV}$. This completes the proof.

Now we are ready to present the theorem about the finite representation of a signed-vector measure.

**Theorem 9** *Suppose that $\theta$ is the parameter with a bounded domain $\Theta \subseteq \mathbb{R}^p$ and $\phi(\mathbf{X}, \theta) : \mathbb{R}^{N \times d} \times \Theta \to \mathbb{R}^N$ is an embedding of the parameter into the feature space. Consider the following optimization problem*

$$\min \|\boldsymbol{\mu}\|_{TV}, \ s.t. \int_{\Theta} \phi(X, \theta) d\boldsymbol{\mu}(\theta) = Y. \tag{109}$$

*Assume that an optimal solution to (109) exists. Then, there exists an optimal solution $\hat{\mu}$ supported on at most $KN + 1$ features in $\Theta$.*

PROOF  Let $\hat{\boldsymbol{\mu}}$ be an optimal solution to (109). We can define a measure $\hat{P}$ on $\mathbb{R}^N$ as the push-forward of $\hat{\boldsymbol{\mu}}$ by $\hat{P}(B) = \hat{\boldsymbol{\mu}}(\{\theta | \phi(X, \theta) \in B\})$. Denote $D = \{\phi(X, \theta) | \theta \in \Theta\}$. We note that $\hat{P}$ is supported on $D$ and $D$ is bounded. By applying Theorem (8) to the set $D$ and the measure $\hat{P}$, we can find a measure $Q$ whose support is a finite subset of $D$, $\{z_1, \ldots, z_k\}$ with $k \leq 2K(n+1)$. For each $z_i \in D$, we can find $\theta_i$ such that $\phi(X, \theta_i) = z_i$. Then, $\tilde{\boldsymbol{\mu}} = \sum_{i=1}^{k} \delta(\theta - \theta_i) dQ(z_i)$ is an optimal solution to (109) with at most $2C(N + 1)$ features and $\|\tilde{\boldsymbol{\mu}}\|_{TV} = \|\boldsymbol{\mu}\|_{TV}$. Here $\delta(\cdot)$ is the Dirac delta measure.

