# OpenReview forum: "Parallel Deep Neural Networks Have Zero Duality Gap"
_ICLR.cc/2022/Conference — ICLR 2022 Submitted_

### Official Review · Reviewer_Pya1 · 2021-10-24

**Correctness:** 2
**Technical Novelty And Significance:** 2
**Empirical Novelty And Significance:** 1
**Recommendation:** 5
**Confidence:** 4

**Main Review:**

Strengths.

    1. The claims are very interesting.
    2. The message delivered in the introduction is clear.

Weakness.

    1. The mathematical presentation is confusing and full of typos. For example:
           1.1. In (1) its W_2, not w_2, this happens many times.
           1.2. In the first formula of section 2, A_{\ell-1} is left-multiplied by W_l. It should be right-multiplied. This happens many times.
           1.3. In the definition of parallel networks, its A_{0,j}, not A_{0}^{j}.
           1.4. Problem (7) has a typo.

    2. There seems to be very few technical novelty. Most of the paper is about the standard step of calculating the dual problem, the rescaling technique is also quite standard and well-known (e.g., in SVM), and was used in (Ergen & Pilanci, 2020b) for neural networks. Please clarify any technical novelty compared to prior works.

    3. There are no experiments that validate the theorems, while the claimed theorems lead to at least the following suspection. Specifically, the main result, Theorem 1, has two closed-form expressions for optimal values of P_{lin}(t) and D_{lin}(t) respectively, and in its proof (page 16) the authors claim that D_{lin}(t) is "strictly" smaller than P_{lin}(t). However, the two terms are obviously equal if X^{dagger}Y has its all singular values equal to 1. This means something wrong in the proof or in the statement of Theorem 1.

    4. Please provide some discussion on m^* of Proposition 1. It seems important to bound it.

    5. Assume Proposition 2 is correct. Further assume that we have ample and general enough data samples (so that X is a tall matrix of full rank). Then I could conclude that the optimal value of (7) is directly given by the optimal value of Proposition 2 where W is replaced by X^dagger Y. As a result, much of the development of the paper becomes pointless as the optimal value of (7), and thus an optimal solution, is already known. In this situation, why do we care about whether the duality gap is zero or not?

Questions: Please shed some light on proof techniques for deep linear networks and deep parallel networks as they are tightly related. My feeling is that the results for deep parallel networks are natural consequences of those for deep linear networks. Is that correct?


**Summary Of The Paper:**

Using duality theory, this paper studies the duality gap between prime and dual optimization problems for deep linear (or relu) networks that have sequential or parallel structure. The major claim is that, assuming the networks is sufficiently over-parameterized, linear networks of depths at least 3 have nonzero duality gap, while strong duality holds for deep parallel networks. Similar conclusions are given for relu networks.

**Summary Of The Review:**

The above concerns on mathematical clarity, technical novelty, technical correctness, and practical vadility lead me to form the opnion of reject. However, I am not an expert in this field of "theory of training neural networks", so I am looking forward to rebuttals from the authors, as well as comments from other reviewers and the area chair.

---

> ### Author Response · Authors · 2021-11-17
> **Response to Reviewer Pya1**
>
> Thanks for your careful reading and constructive comments. We will include the suggestions in the final version, as well as the pointed references. In the remainder, we want to address the main points raised in the reviews.
>
> 1. We fixed the typos in the revision.
>
> 1.1 The minimization should be with respect to $W_1$ and $w_2$. Here we use the lowercase letter to emphasize that this is a scalar-output NN.
>
> 1.2 It should be $A_{l-1}W_l$.
>
> 1.3 It should be $A_{0,j}$.
>
> 1.4 Yes, the subscript shall be l instead of L.
>
> 2. We agree that the rescaling argument is well-known in the literature. Our main contribution is to derive the convex dual problem for multi-layer NNs with vector outputs and demonstrate the duality gap for standard NNs and parallel NNs, as detailed in Table 1. We are not aware of any existing results for strong duality in deep ReLU networks (deeper than three layers).
>
> 3. Thanks for pointing this out. We agree that the duality gap is zero when the singular values of $X^{\dagger}Y$ are the same. In general, the duality gap is non-zero since the singular values of $X^{\dagger}Y$ will not be all the same. We emphasized this after the statement of Theorem 1 in the revision.
>
> 4. Indeed we discuss the upper bound of $m^*$ based on Caratheodory's theorem and finite representation in Appendix E. We added these discussions to the revised version of the paper.
>
> 5. Note that the main purpose of presenting results on deep linear networks is that there is an interesting difference between parallel and non-parallel deep networks. Our analysis illustrates that even for deep linear networks, the duality gap is non-zero for standard NNs but it is zero for parallel NNs. It is important to note that zero-duality gap implies that the problem is essentially convex.
>
> Moreover, for the deep linear network, the optimal solution is $X^{\dagger} Y$ only when loss function is the squared loss. For other loss functions, the optimal deep linear network for the problem described in eq. (20) can be different, which is still covered by our theory.
>
> On the other hand, we acknowledge that a deep linear network is still an over-parameterized linear model. However, the minimum norm condition induces a regularization on the weights. For standard networks, the $\ell_2$ regularization on the layer weights indeed yield a Schatten quasi $2/L$-norm regularization on the weight matrix of the linear model.  For parallel networks, the regularization is just the nuclear norm, which provides a different model.
>
> Regarding the last question:
>
> Yes, the results for deep linear NNs serve as a starting point for us to show that the duality gap can be non-zero for deep ReLU NNs and they motivate us to consider the parallel structure for deep architectures in order to make the duality gap zero. Still, our results for deep ReLU NNs are more challenging to prove because the feasible set of the dual problem is more complicated to characterize compared to the one for the deep linear NN.

---

> > ### Comment · Reviewer_Pya1 · 2021-11-23
> > **Thanks for the response**
> >
> > Much of my concerns have been addressed in the rebuttal. I believe the paper is now much stronger. For example, a message there is that there is a closed-form solution to training deep linear networks (with squared loss). For this, I have raised my score to borderline.

---

### Official Review · Reviewer_eU6M · 2021-10-31

**Correctness:** 4
**Technical Novelty And Significance:** 3
**Empirical Novelty And Significance:** Not applicable
**Recommendation:** 6
**Confidence:** 3

**Main Review:**

Strength:
1. Analyzing the duality gap in neural network is an important research question both theoretically and empirically.
2. The paper is well written and very good to follow.
3. Showing the duality gap is non-zero for three-layer ReLU networks is important.

Weakness:
1. The author proves that the duality gap is zero when the neural network has parallel architecture, but I find there are little discussion behind this setting, i.e. why the parallel architecture can help closing the duality gap?
2. Is the parallel architecture setting practical? Has it been used empirically?
3. There is no numerically experiments supporting the theoretical result.
4. In general, even when the duality gap is zero and we can solve the dual problem, how can we recover the primal solution?

**Summary Of The Paper:**

In this paper, the author studies the duality gap in the neural network training problem. They show that in general (deep linear networks and three-layer ReLU networks), the duality gap is not zero. However, when restricting the parallel architecture, the authors show that the duality gap is indeed zero.

**Summary Of The Review:**

I found the manuscript to be clearly written and technically sound. Although it has some weakness points, I think it still worth a publication.

---

> ### Author Response · Authors · 2021-11-17
> **Response to Reviewer eU6M**
>
> Thanks for your careful reading and constructive comments. We will include the suggestions in the final version, as well as the pointed references. In the remainder, we want to address the main points raised in the reviews.
>
> 1. We will add an intuitive explanation in the final version based on convex duality theory. Namely, the bi-dual convex program, i.e., the dual of the convex dual, corresponding to the non-convex minimal norm NN problem in fact reduces to the parallel NN problem. In this sense, parallel NNs can be thought as the convex hull of non-parallel NNs. As there is no duality gap between the bi-dual problem and the dual problem, it is natural to study the parallel NNs, which are at least as expressive as non-parallel ones. We will elaborate more the motivation of studying parallel NNs in the final paper.
>
> 2. Yes. We discuss the practical applications of parallel NNs in the paragraph above subsection 1.1, where we list several references that show empirical improvements over standard networks.
>
> 3. Currently, the scope of our theory is limited to the minimal norm problems, where the regularization parameter in the regularized training problem goes to zero. However, solving the minimal norm problem with the equality constraints on the output of the network is challenging, although it can be numerically approximated by picking a very small regularization parameter. We will comment on this issue in the final paper. Still, we believe that our results can be extended to regularized problems.
>
> 4. Given an optimal dual solution, we can recover the primal solution by finding the activate constraints based on the dual solution. For example, in eq (30), the matrices $W_1,...,W_{L-1},w_{L-1}$ that are active in the constraint at optimum are the optimal parameters in each branch for the primal parallel NN problem. Alternatively, one can solve the convex bi-dual problems, which are the duals of the dual problems.

---

### Official Review · Reviewer_c3BS · 2021-11-01

**Correctness:** 2
**Technical Novelty And Significance:** 3
**Empirical Novelty And Significance:** Not applicable
**Recommendation:** 5
**Confidence:** 4

**Main Review:**

[Contributions]

The analysis of the loss landscape of deep neural networks is an important topic to understand the global convergence property of optimization. Recently, the dual formulation of two-layer neural networks has been studied and the strong duality has been shown by the series of [Ergen and Pilanci] works and [Zhang et al. (2019)]. A natural question is to extend these results to deep models and this study answer the question. In particular, the strong duality theorem for the parallel deep ReLU model is quite interesting.


[Weaknesses]

I basically like the theoretical results in this paper, but I think the quality should be much improved and the paper is not ready for publication because of typos, ambiguity, and the lack of proof.  For the detail see below.

- I could not find the proof of the first part of Proposition 4 and 6. These are important statements that make the equivalence between (16) and (17), (28) and (29).
- Notations (Subsection 1.2): The notation $W_l \in \mathbb{R}^{m_1\times m_2}$ is somewhat confusing because $W_l$ represents the parameter of $l$-th layer in the main text.
- Eq. (1): $W_2$ → $w_2$. Moreover, there are many similar typos with the wrong uppercase and lowercase letters in the paper.
- Section2:
    - The formulation of standard networks: $W_l A_{l-1}$ → $A_{l-1}W_l$.
    - Inconsistent conditions: $m_l \geq \max\\{d,K\\}$, ($l \in [L-1]$), and $m_{L-1}=1$.
    - Lack of specification for the type of $A_{L-1}$ and $W_L$.
- Eq. (7): $\|W_L\|_F^2$ → $ \|  W_i \|_F^2$ ?
- Eq. (9): Notation $\forall$ is in the wong place ($\forall \| W_i \|_F$ → $\forall i \in [L-2]$).


[Additional references]

It is interesting that bi-dual problems become learning problems of mean field neural networks with TV-norm regularization and additional constraints (e.g., (41), (79), (97)). Hence, I think the strong duality theorem for parallel models is a reasonable consequence because parallel models are essentially neural networks in the (finite-width) mean field regime. There are many studies on the mean field regime. For instance,

- A. Nitanda and T. Suzuki. Stochastic particle gradient descent for infinite ensembles. arXiv, 2017.
- S. Mei, A. Montanari, and P. M. Nguyen. A mean field view of the landscape of two-layer neural networks. PNAS, 2018.
- L. Chizat and F. Bach. On the global convergence of gradient descent for over-parameterized
models using optimal transport. NeurIPS, 2018.
- S. Mei, T. Misiakiewicz, and A. Montanari. Mean-field theory of two-layers neural networks:
dimension-free bounds and kernel limit. COLT, 2019.
- G. M. Rotskoff, S. Jelassi,  J. Bruna, and E. Vanden-Eijnden. Global convergence of neuron
birth-death dynamics. ICML, 2019.
- J. Sirignano and K. Spiliopoulos. Mean field analysis of neural networks: A central limit theorem. Stochastic Processes and their Applications, 2020.
- S. Akiyama and T. Suzuki. On learnability via gradient method for two-layer relu neural networks in teacher-student setting. ICML, 2021.
- L. Chizat. Sparse optimization on measures with over-parameterized gradient descent. Mathematical Programming, 2021.
- A. Nitanda, D. Wu, and T. Suzuki. Particle dual averaging: Optimization of mean field neural
networks with global convergence rate analysis. NeurIPS, 2021.

**Summary Of The Paper:**

The topic of this paper is the duality theorem for deep neural networks with linear and ReLU activation functions. In particular, this study shows that the strong duality (i.e., zero duality gap) does not hold for the standard deep models but it holds for parallel deep neural networks which are ensembles of networks with an appropriate regularization. These results are certainly useful because it enables global optimization of parallel deep models via the convex dual programs.

**Summary Of The Review:**

The statements of main theorems are interesting and important, but the quality of the paper should be much improved. See the main review for detail.

---

> ### Author Response · Authors · 2021-11-17
> **Response to Reviewer c3BS**
>
> Thanks for your careful reading and constructive comments. We included the missing derivations in the revision, as well as the suggested references. In the remainder, we want to address the main points raised in the reviews.
>
> 1. The derivations of eq. (17) and eq. (28) follow from the rescaling argument and the fact that the arithmetic and geometric mean (AM-GM) inequality holds with equality. We included the detailed derivation in the revision.
>
> 2. We changed the notation to $W_l\in \mathbb{R}^{m_{l-1}\times m_{l}}$.
>
> 3. The minimization is with respect to $W_1$ and $w_2$. Here we use the lowercase letter to emphasize that this is a scalar-output NN.
>
> 4. This is a typo. It should be $A_{l-1}W_l$, which is corrected in the revised version. For the parallel NN, we require that each branch is a scalar output neural network so the condition is $m_{l}\geq \max_{d,K}$ for $l\in[L-2]$ and $m_{L-1}=1$. We added the specifications such that $A_{L-1}\in \mathbb{R}^{N\times m}$ and $W_L\in\mathbb{R}^{m\times K}$.
>
> 5. Yes, the subscript should be $l$ instead of $L$. This is corrected in the revision.
>
> 6. Yes. We corrected this typo in the revision.
>
> We will include these references from the mean-field approximations literature for NNs.

---

> > ### Comment · Reviewer_c3BS · 2021-11-23
> > **Thanks for the response.**
> >
> > I have checked the revised version of the paper. Thanks for completing the proof of Proposition 4 and 6, and correcting typos.
> > I believe that a discussion about the connection with mean field regime will be helpful in increasing the importance of the paper because this regime can explain the feature learning aspect of this parallel architecture. Regarding this, the following paper might be useful.
> >
> > Greg Yang and Edward J. Hu. Tensor programs IV: Feature learning in infinite-width neural networks. ICML, 2021.
> >
> > I hope this discussion will be included in a future version. I would like to increase the score: 3 -> 5.

---

### Official Review · Reviewer_3piH · 2021-11-08

**Correctness:** 3
**Technical Novelty And Significance:** 3
**Empirical Novelty And Significance:** Not applicable
**Recommendation:** 5
**Confidence:** 4

**Main Review:**

The paper tackles an interesting and I believe important problem in NN optimization. Although the results are limited to minimum norm training of over-parametrized NN, the theoretical analysis may be useful in future development and analysis of deep NN optimizations. The paper has some shortcomings and some parts of the analysis were not clear, such as

1. As a theoretical paper, I expected to see more precise statements. For example, the paper claims that the duality gap is zero for parallel NN, but it seems that this is true only for NNs with enough number of parallel branches.
2. Do the authors define parallel NNs such that the output of each branch is a scalar?
3. Some equations and derivations are not clear or obvious, e.g., proof of proposition 4, why equation (64) is equivalent to (16)? Why the power $L$ is dropped? Is deriving equation (22) similar to the ones in the references ? In the proof of proposition 6, it is not clear how the power $L$ (in equation (28), $\\|W_L\\|^L$) is dropped and the optimizaiton become over $\ell_2$ norm of the rows of $W_L$.
4. In the paragraph after equation (5), the authors claim that "The rescaling does not change the solution to (4).". Although by appropriate scaling $W_1$ and $W_2$ can still satisfy the constraint, but as the value of the objective function changes, the statement seems  inaccurate. Can the authors explain?
5. In section 4.1, Theorem 3, the authors assumed that data is rank 1. Does it simply imply that data samples are simply given by a scaling of a base vector, i.e., $x_i = s_i a$, for scalar $s_i$?
6. Theorem 5, is there any constraint on the number of branches or $m$?

There are some minor typos in the equations, such as
- Section 2, first equation, $A_l=\Phi(W_lA_{l-1})$ should be $A_l=\Phi(A_{l-1}W_l)$. Similarly, for the second set of equations in that section.
- Is summation in equation (7) over $\\|W_i\\|^2$ rather than fixed $\\|W_L\\|$?
- First sentence section 4.2, "there is no duality gap for arbitrary deep ReLU networks", seems "parallel with large enough number of branches" should be added to make the statement accurate.


**Summary Of The Paper:**

The authors theoretically analyzed the dual of minimum norm optimization of over-parameterized deep neural networks. Specifically, they focused on deep neural networks with linear and ReLU activation functions, as well as parallel architectures where the output of each branch is a scalar. First, the authors considered the linear NNs and showed that the duality gap of general deep NN is non-zero, but for parallel NN with enough number of parallel branches, that would be zero. Next, they considered neural networks with ReLU activation functions and derived the duality gap for standard 3 layers neural networks. However, for an arbitrary deep parallel NN with ReLU activation, the duality gap would be zero if the L-th power of Frobenius norms of the weights is used as the regularizer.

**Summary Of The Review:**

The ideas and analysis of the paper to find the duality gap in training deep neural networks and when it achieves zero (at least for over-parametrized deep NNs) can be useful in understanding the optimization and training deep NNs and their properties. The paper adds non-trivial and solid results to the previous works in this domain, esp. works by Ergen and Pilanci. However, the writing and presentation of the paper falls short from the expectations.

---

> ### Author Response · Authors · 2021-11-17
> **Response to Reviewer 3piH**
>
> Thanks for your careful reading and constructive comments. We will include the suggestions in the final version. In the remainder, we want to address the main points raised in the reviews.
>
> Response to the main concerns:
> 1.  We will modify the statement to `the duality gap is zero for parallel NNs with enough number of branches'. However, we also have an upper-bound on the number of branches for our result to hold. We will emphasize this in the revision.
>
> 2. Yes. We define the parallel NNs in the beginning of section 2 such that they have scalar output from each branch.
>
> 3. Thanks for pointing these out. We added explanations in the revision. The Lagrangian function defined in eq. (64) is defined w.r.t. eq. (17). The reformulation of eq. (16) into eq. (17) and the reason why the power L is dropped follow from the rescaling argument and the fact that the arithmetic and geometric mean (AM-GM) inequality is achieved.
>
> Deriving eq. (22) parallels the derivation of Lemma 1.1 in [1].
>
> The reformulation into eq. (28) is analogous to the reformulation into eq. (17).
>
> We included a detailed derivation of eq. (17), eq. (22) and eq. (28) in the revision.
>
> 4. This statement in the literature review part is explained in [2] (e.g., in the derivation of Lemma 2 in [2]).
>
> 5. Yes. Note that the rank 1 case is investigated as a simple special case.
>
> 6. We require that the number of branches is greater than a threshold, i.g., $m\geq m^*$, where $m^*$ is upper bounded by $KN+1$. Here $K$ is the number of classes and $N$ is the number of samples.
>
> Response to the minor typos:
>
> 1. Thanks for pointing it out. We correct it in the revision.
>
> 2. Yes, the subscript should be l instead of L.
>
> 3. Yes. We changed it to `with large enough number of branches' in the revision.
>
> [1] Tolga Ergen and Mert Pilanci, Revealing the Structure of Deep Neural Networks via Convex Duality.
>
> [2] Mert Pilanci and Tolga Ergen, Neural Networks are Convex Regularizers: Exact Polynomial-time Convex Optimization Formulations for Two-Layer Networks.

---

### Decision · Program_Chairs · 2022-01-20

**Decision:**

Reject

**Comment:**

This is an interesting paper which further extends the duality theory of deep networks.  Unfortunately, reviewers had many concerns about, presentation, technical details, and missing prior work.  I will add that a large volume of relevant implicit bias work (e.g., in the setting of deep linear networks, mirroring Proposition 2) is completely uncited (e.g., works by Arora et al., Soudry et al., Ji et al.), despite being earlier than many of the works which are currently cited.  As such, I urge the authors to continue in their valuable line of work, taking into consideration all of these points and also the reviewer comments.

Separately, I note that there is a violation of the blind policy in the current revision: grant information was included.  The PC decide this was a minor violation and should not affect the review process, however their decision could have easily been otherwise.  I urge the authors to be exceptionally careful with such issues in the future.